# Aircraft ice-nucleating particle and aerosol composition measurements in the Western North American Arctic

Alberto Sanchez-Marroquin[1], Sarah L. Barr [1], Ian T. Burke[1], James B. McQuaid[1], Benjamin J. Murray[1] (*)

[1]School of Earth and Environment, University of Leeds, Woodhouse Lane, Leeds, LS2 9JT, UK

(*) Corresponding author: b.j.murray@leeds.ac.uk

Knowledge of the temperature dependent concentration of ice-nucleating particles (INPs) is crucial to understanding the properties of mixed-phase clouds. However, the sources, transport and removal of INPs around the globe, and particularly in the Arctic region, are poorly understood. In the Arctic winter and spring, when many local sources are covered by ice and snow, it is not clear which INP types are important. In this study, we present a new dataset of aircraft-based immersion mode INP measurements and aerosol size-resolved composition in the Western North American Arctic from the 11[th] – 21[st] March 2018. Aerosol samples were collected between ~70 and 500 m above the surface on filters that were analysed using both a freezing droplet-based assay and Scanning Electron Microscopy with Energy Dispersive Spectroscopy (SEM-EDS). The measured INP concentrations were at or close to the limit of detection, with concentrations at -20°C of 1 L$^{-1}$ or below. The size-resolved composition measurements indicates that the aerosol concentrations were low, dominated mostly by sea spray aerosol and mineral dust. Further analysis shows that mineral dust is important for the ice-nucleating properties of our samples, dominating over the sea spray aerosol particles in the four cases we analysed, suggesting that mineral dust is a relevant source of INPs in the Alaskan springtime Arctic. Furthermore, the INP concentrations are more consistent with fertile soil dusts that have an ice active biological component than what would be expected for the ice-active mineral K-feldspar alone. While we cannot rule out local high latitude sources of dust, the relatively small size of the mineral dust implies that the dust was from distant sources.

## 1. Introduction

Clouds containing both supercooled liquid water and ice are known as mixed-phase clouds and they reflect a substantial amount of the incoming solar shortwave radiation that reaches the Earth (Boucher, 2013). The lifetime, as well as the amount of radiation that these clouds reflect, is strongly affected by the partitioning between liquid and ice (Storelvmo et al., 2015). When above temperatures required for homogeneous freezing (below ~-35°C), ice formation in mixed-phase clouds is initiated by the presence of a small fraction of the aerosol particles known as ice-nucleating particles (INPs) (Murray et al., 2012). Once ice crystals nucleate, they can grow more rapidly than liquid cloud droplets since ice has a lower equilibrium vapour pressure than supercooled water. This process can lead to the precipitation of the ice crystals, removing liquid water from a cloud (Korolev et al., 2017;Vergara-Temprado et al., 2018;Hawker et al., 2021). Ice-related processes in mixed-phase clouds such as the primary production of ice and the link to INP concentration are commonly oversimplified in climate models, which contributes to large discrepancies in the amount of water and ice that the models simulate (Komurcu et al., 2014;McCoy et al., 2016;McCoy et al., 2018). The difficulty of properly representing the current

water and ice mixing state of these clouds is responsible for the large uncertainty of the cloud-phase
feedback (Storelvmo et al., 2015).
As the atmosphere warms, mixed-phase clouds will contain more supercooled water leading to a
reduction in shortwave radiation reaching the surface, but also decrease the outgoing longwave radiation
flux (Ceppi et al., 2017;Murray et al., 2021). Hence, mixed-phase mid- to high-latitude clouds over the
ocean have a negative feedback (Tan et al., 2016), whereas clouds over high albedo ice or snow covered
surfaces have a positive feedback (Tan et al., 2016). The strength of these feedbacks depends on the
balance between ice and supercooled water in these clouds both in the present and future climate. Hence,
better understanding the sources and concentrations of atmospheric INPs, particularly at the mid- to
high-latitudes could help to reduce the uncertainty associated with cloud feedbacks.
Only a small fraction of aerosol particles have the potential to become INPs. Transported dust from the
deserts is one of the most important sources of worldwide atmospheric INPs, especially at temperatures
below -15 °C (Hoose and Mohler, 2012;Vergara-Temprado et al., 2017;Kanji et al., 2017). Given the
fact that substantial amounts of dust are transported from the deserts to the Arctic (Fan, 2013;Huang et
al., 2015;Francis et al., 2018), this dust could contribute to the INP population of the region (Irish et al.,
2019;Yun et al., 2022). Additionally, local sources of high-latitude dust are known to contribute to the
dust budget in the Arctic (Bullard et al., 2016;Groot Zwaaftink et al., 2016;Meinander et al., 2022;Shi
et al., 2021). Some of these sources of high-latitude dust have been found to contribute to the Arctic
INP population (Tobo et al., 2019;Sanchez-Marroquin et al., 2020;Si et al., 2019). A fraction the INP
in the Arctic are also of biogenic origin (Wex et al., 2019;Porter et al., 2022), some of which may be
associated with biogenic material in sea spray and some of which may be from terrestrial sources
(Wilson et al., 2015;DeMott et al., 2016;Vergara-Temprado et al., 2017;Irish et al., 2017;McCluskey et
al., 2018;Bigg and Leck, 2001;Creamean et al., 2019;Hartmann et al., 2020;Creamean et al., 2020).
Biogenic material attached to dust particles could be an important part of these terrestrial INPs
(O'Sullivan et al., 2014;O'Sullivan et al., 2015;Tobo et al., 2019). Other types of aerosol particles such
as volcanic ash or biomass burning particles could also contribute to the INP population in the Arctic
(Prenni et al., 2009).
The available literature data indicates that the INP concentrations in the Arctic are highly variable,
depending on the season and location (Murray et al., 2021). Using samples from land-based sites around
the Arctic collected over several years, Wex et al. (2019) found that Arctic INP concentrations reach a
minimum during winter, but they increase through spring and reach a maximum around the summer,
suggesting that concentrations are highest when the transport of aerosol from the low latitudes is at its
weakest (the summer). Similarly, year-round measurements in the central Arctic indicate peak
concentrations in the summer months of 2020 (Creamean et al., 2022). Creamean et al. (2022) suggested
that local Arctic marine sources might contribute to the elevated INP populations in the summer. Porter
et al. (2022) also found elevated summertime INP concentration, during August 2018, while in the pack
ice near the North Pole. However, in contrast to Creamean et al. (2022), Porter et al. (2022) concluded
that these very active INPs were associated with air masses originating from lower latitude ice-free
regions along the Russian coast, whereas air masses that had spent the preceding week or so over ice-
covered surfaces (in the central Arctic pack ice) had very low INP particle concentrations . The central
Arctic in 2018 and 2020 appear to be rather different, with Porter et al. (2022) reporting up to 2 L$^{-1}$ at -
15°C in 2018, whereas Creamean et al. (2022) reported two orders of magnitude lower peak INP
concentrations in 2020. Hence, there may simply be a great deal of variability and the contrasting
conclusions between Porter et al. (2022) and Creamean et al. (2022) may be appropriate for their
respective study periods. Creamean et al. (2018) found a similar trend in INP concentrations to Wex et
al. (2019) over spring, with coarse particles being responsible for the higher INP concentration event.
However, a recent study did not find strong seasonality of Arctic INPs at Ny-Ålesund, although these
measurements were limited to being between April and August 2018 (Rinaldi et al., 2021). Furthermore,
there have been very few INP measurements from aircraft. Given there are strong aerosol sinks in the
boundary layer, whereas the air above the boundary layer can be stratified with corresponding long
aerosol lifetimes (Carslaw, 2022), vertical measurements are required.  Hartmann et al. (2020) report
INP spectra for late March and early April north of 80° over the Fram Strait and Arctic Ocean and report
that the highest INP concentrations ($2 \times 10^{-2}$ $L^{-1}$ at -15°C) correspond to the boundary layer, indicating
a local marine source even though the region was mostly ice covered. Overall, the picture of INP
concentrations in the Arctic is that of high variability, both spatially and temporal (on days to years
timescales), with the potential for high variability in local sources, transport from lower latitudes as
well as in local INP sinks.
In this paper, we present a set of immersion mode INP and aerosol size-resolved composition
measurements carried out in the Western North American Arctic during March 2018 using an aircraft.
INP measurements were combined with aerosol characteristics determined using SEM-EDS to indicate
the types of INPs that are were most important during this campaign.

## 2. Sampling location and methods

Aerosol particles were sampled from the UK's BAe-146 FAAM atmospheric research aircraft during the Measurements of Arctic Cloud, Snow and Sea Ice in the Marginal Ice ZonE (MACSSIMIZE) campaign, based in Fairbanks, Alaska (US) in March 2018. The majority of the measurements were carried out close to the northern coast of Alaska and the Canadian territory of Yukon, both over land and over the Arctic Ocean, as shown in Fig. 1, where the approximated midpoint of the filter sampling run locations are shown with a star. Measurements were carried out at altitudes between 40 and 600 m above sea level, as detailed in Table 1 along with other pertinent information. Some filters were collected in a single run on a constant heading and height, while others were collected over several runs, with the filters system mostly closed during turns between the runs and altitude changes, although this was not possible for all filters. Filters were collected over 9 to 36 minutes, which at a science speed of ~360 km hr$^{-1}$ corresponded to a horizontal distance of between ~50 and ~200 km. The sampling locations as well as sampling altitudes are shown in Fig. 1. All the sampling was done outside of cloud and precipitation.

Aerosol particles were collected using the filter inlet system on board of the FAAM BAe-146, which has been characterised by Sanchez-Marroquin et al. (2019). Briefly, this inlet is located outside the skin of the FAAM BAe-146 and brings aerosol particles to a filter located inside the cabin with a 45$^{o}$ angle bend. The sampling occurs in sub-isokinetic conditions, which enhances coarse mode aerosol particles. Sampling efficiency for particles with diameters above 20 µm becomes very small due to inertial losses in the system (at the bend). No treatment (heat or drying) is applied to the sampled airmass, although the cabin was warmer than the ambient air in this campaign and hence the RH of air passing through the inlet system once inside the aircraft is therefore very low. The system allowed us to collect two aerosol samples in parallel: one on a polycarbonate filter (Whatman Nuclepore polycarbonate track-etched filters, 47 mm diameter with a pore size of 0.4 µm) and one on a Teflon filter (Sartorius polytetrafluoroethylene, 47 mm diameter with a pore size of 0.45 µm). For these filter types, the particle collection efficiency is likely to be close to 100 % for the relevant size ranges, as discussed in Sanchez-Marroquin et al. (2019) using the data of Soo et al. (2016) and Lindsley (2016).

The ice-nucleating particle assay was conducted in a temporary laboratory set up in a hotel room near the aircraft base in Fairbanks, Alaska, with minimum time between sampling and analysis. Most filters were analysed a matter of hours after collection, however where this was not possible they were stored at ~ -18 $^{o}$C for a few days prior to analysis. This approach has a number of advantages compared to the commonly used strategy of bringing filters back to a laboratory for latter analysis. Firstly, analysis of field blanks can reveal sources of contamination that can be reduced by making adjustments to the experimental protocol; secondly, we can try to adjust the sampling methodology (such as sampling time) to fit the INP concentration and thirdly, we can minimise storage and transport of filters thus reducing potential biases. Teflon filters were used to perform a droplet-on-filter freezing assay to quantify the INP concentration, as described in detail in Price et al. (2018) and also used by Sanchez-Marroquin et al. (2020) and (Sanchez-Marroquin et al., 2021). The technique was first described by Schnell (1982) and our version of this assay makes use of the Asymptote EF600 Stirling cooler described in Whale et al. (2015). For the present study we pipetted 2 µL pure water droplets on top of each filter that had been exposed to aerosol particles (or handling blanks). On average, we pipetted 54 (with and standard deviation of 5) droplets per filter. The filters were placed on top of a cold stage, within a chamber that is flushed with dry nitrogen gas to prevent water condensation, that is cooled at a constant rate of 1 $^{o}$C min$^{-1}$ until temperatures of ~-35 $^{o}$C. Droplet freezing was recorded and the resulting videos were manually analysed to determine the fraction of droplets frozen at each temperature and then the INP concentration. At least one handling blank experiment was performed for every flight. Handling blank filters were prepared and transported in the same way as the measurement filters including loading the filters into the sampling system on the aircraft and briefly opening (for a second or so) and closing the inlet valves that allow air to pass through the filters. Hence, the handling blank

should provide information on sources of contamination throughout the handling of the filter. A
disadvantage of the droplet-on-filter technique is that each sample can only be analysed once, which
makes it incompatible with standard heat tests such as the ones described by Daily et al. (2022).
However, the great advantage of the droplet-on-filter technique over techniques where particles are
washed off a filter into a volume of water is that it is around 20 times more sensitive than a typical
wash-off assay employing 1 μL droplets (depending on the details of the freezing assays). This
enhanced sensitivity is very important given that aerosol sampling durations are typically only a few
tens of minutes long.
The droplet fraction frozen (the fraction of droplet that were frozen as a function of temperature)
produced by our samples, along with those produced by the handling blank filters, is shown in Fig 2a.
While the fraction frozen for the sample filters were generally shifted to warmer temperatures than the
handling blanks, many of the samples overlapped with the range defined by the handling blanks. Hence
it was necessary to account for influence of the background from the measurements. The background
subtraction procedure and the INP concentration calculations are detailed in Appendix A. Briefly, we
converted our cumulative fraction frozen values for the samples and handling blanks into the differential
INP spectrum, $k(T)$, in units of INP per unit temperature (Vali, 1971;Vali, 2019). $k$ is the number of
INP that become active in a temperature interval. This allowed us to define a limit of detection then
apply a criterion to separate samples that show a significant signal above this from the ones that do not.
Data points whose error bars did not overlap with the error bars associated to the handling blank were
considered to be above the limit of detection. The error bars of the differential concentrations of the
samples represent a confidence level of 68 % while the error bars of the background represents the
standard deviation of all the measured handling blanks. Background-subtraction was applied to data
points above the limit of detection($k_{sample} - k_{background}$) using a similar approach to Vali (2019). The
cumulative INP spectrum, the common way of presenting INP data, was then derived using the
background corrected values of $k$.
A subset of the polycarbonate filters was analysed using Scanning Electron Microscopy with Energy
Dispersive Spectroscopy (SEM-EDS) to study aerosol size-resolved composition. The analysis was
carried out in the Leeds Electron Microscopy And Spectroscopy centre (LEMAS), at the University of
Leeds. Filters were transported to the University of Leeds and then stored at ~-18 °C until its analysis.
This technique can be used to obtain the morphological and chemical properties of individual aerosol
particles within the sample. The subset of samples was coated with a 30 nm layer of Iridium and the
SEM-EDS analysis was performed using an accelerating voltage of 20 KeV. The scanning and
acquisition of EDS spectrums is done using a semi-automatic method with the Aztec Feature Software
by Oxford Instruments. Our method captures the morphology and chemical signature of particles down
to 0.2 or 0.3 μm, depending on the sample. Particles are detected based on their contrast in the secondary
electron images, although some artefacts were removed manually. Each particle is then classified into
a defined composition category based on its chemical composition. The morphological and composition
category of each particles is used to obtain statistics about the size-resolved composition of the aerosol
samples. A more detailed description of the technique can be found in Sanchez-Marroquin et al. (2019).
In parallel with the filters sampling, we make use of FAAM's underwing optical particle counters. One
of these counters is the Passive Cavity Aerosol Spectrometer Probe 100-X (PCASP), manufactured by
Particle Measurement Systems, and measures aerosol particles in the 0.1 to ~3 μm range. The second
counter is the Cloud Droplet Probe (CDP) by Droplet Measurement Technologies and it measures
aerosol particles and droplets with sizes from ~3 to 50 μm. A detailed description of these instruments
and its calibration and can be found in Rosenberg et al. (2012).
The Hybrid Single-Particle Lagrangian Integrated Trajectory (HYSPLIT) model was used to calculate
five day back trajectories of sampled air masses (Stein et al., 2015;Rolph et al., 2017) and shown in
Sect S1. The back trajectories show that in many cases air masses remained near or over Alaska and

northern Canada before sampling. However, the backtrajectories corresponding to the C085 flight arrived mostly from the south west. Most of the trajectories suggest that the air mostly stayed at altitudes below 1000 m above sea level in the five days prior to sampling. At the time of sampling, most of the sea and land surfaces were covered by sea ice or snow (Fig. 1), which most likely suppressed any local aerosol sources. However, local sources of marine aerosol particles may still occur due to open leads (May et al., 2016;Kirpes et al., 2019;Chen et al., 2022).

## 3. INP concentrations in the Western North American Arctic

The background corrected cumulative INP concentrations are shown in Fig 2b. Hollow markers indicate measurements consistent with the limit of detection, where the lower error bar goes to zero, while filled markers correspond to a cumulative INP concentration above the limit of detection. Using a 68 % confidence interval, approximately 70 % of the differential spectra binned data was not significantly above the limit of detection and around half of the data points in the cumulative INP spectra shown in Fig. 2b show INP concentrations consistent with zero (i.e. not above the detection limit). The reported INP concentrations are always below 0.1 and 1 $L^{-1}$ at -15 $^{o}$C and -20 $^{o}$C, respectively. However, given the fact that a substantial percentage of the data is not above the detection limit, the real values of some of these samples may be well below these values. A daily, more detailed representation of the INP concentrations is shown in Fig. B3.

INP concentrations across the Arctic vary significantly depending on the time of the year and location (Creamean et al., 2018;Si et al., 2019;Wex et al., 2019). Hence, in order to compare to the pertinent data we show our INP concentrations alongside literature data collected in a similar location and time of the year in Fig. 3 (we restricted the literature datasets from February to April). Some of our reported INP concentrations are above some of the values measured using a droplet freezing assay on filters collected the surface by Creamean et al. (2018) and Wex et al. (2019) as well as filters collected on an aircraft and processed using a dynamic developing chamber at water saturation by Borys (1989). Creamean et al. (2018) reported INP concentrations at -20 $^{o}$C up to 0.01 $L^{-1}$ on the north coast of Alaska in March. Measurements performed by Wex et al. (2019) in a close location (Utqiaġvik) indicate that INP concentrations ranging from ~$10^{-4}$ to $10^{-2}$ $L^{-1}$ at -10 $^{o}$C in March. The more active samples reported by Wex et al. (2019) form a consistent INP spectrum with our more active samples, but unfortunately there is no direct overlap. Borys (1989) reported INP concentrations of 0.001 $L^{-1}$ to 0.3 $L^{-1}$ at -25 $^{o}$C measured from an aircraft at a similar location and time of the year. These values are of course consistent with our samples where we report upper limits, but some of our samples clearly had substantially higher INP concentrations than the range reported by Borys (1989). Hiranuma et al. (2013) also report INP measurements using an airborne continuous flow diffusion chamber (CFDC) during the Indirect and Semi-Direct Aerosol Campaign (ISDAC) in a very similar study region to ours, but in April rather than March. We have only compared our measurements with theirs at water saturation, which happened to be during a relatively high INP period. This INP value of 5.6 ± 3.5 $L^{-1}$ at -22°C is consistent with our highest recorded INP concentrations. Overall, this comparison with measurements in previous years at a similar location and time of year indicates that the INP concentrations are rather variable, ranging over at least three orders of magnitude at -20°C.

Our measurements have also been presented alongside a compilation of INP measurements from across the Arctic carried out throughout the year (Fig 3b). Our dataset is well within the range of literature INP measurements from across the Arctic. Around 50% of our data points were below detection limit (and not shown in Fig. 3), hence we are only able to report INP concentrations when their values are relatively high. The picture that emerges in the Arctic is a region of highly variable INP concentrations. This variability is likely related to a combination of transport from local and remote sources of INP as well as sinks both locally and along those transport routes. This high variability in INP concentrations will affect primary ice production in clouds, with more INP leading to greater ice concentrations that

may or may not be amplified by secondary production processes. Intriguingly, several authors report that greater INP concentrations leads to more ice in Arctic cloud and vice versa (Rogers et al., 2001;Hiranuma et al., 2013).

A handful of measurements of INP have been made from aircraft (Hartmann et al., 2020;Sanchez-Marroquin et al., 2020;Prenni et al., 2009) and it is these measurements that produce many of the highest observed Arctic INP concentrations, rather than those made on the ground. However, aircraft sampling is often limited by the volume of air that can be sampled due to restrictions in flight lengths and other technical limitations. This necessarily biases the results to relatively high INP concentrations. For example, (Rogers et al., 2001) report that 50% of the 10 s averaged data was zero (i.e. below detection limit). Given the Arctic atmosphere is highly stratified, it would be interesting to perform simultaneous measurements at the surface and from an aircraft to explore how or if INP at the surface are related to those higher in the boundary later and those in the free troposphere.

## 4. SEM-EDS size-resolved composition analysis

The equivalent circular diameter size distributions obtained with the SEM-EDS technique were compared with the average size distributions for the same sampling periods measured using the underwing optical particle counters on-board of the FAAM BAe-146. The analysis is shown in Fig. 4 alongside the size-resolved chemical composition of the analysed samples. The number size distribution is multiplied by the fraction of particles in each category and binned to calculate the number size distribution of each category. Then these number size distributions are turned into surface area size distributions and integrated to obtain the surface area of each category, as shown in Table 2.

The analysed samples exhibited low aerosol concentrations relative to other locations where we have used this technique, especially for the coarse mode. In this study, almost no particles above 10 μm were detected, in contrast to samples from around Iceland, the eastern tropical Atlantic and the south east of the United Kingdom analysed using the same or similar technique, where significant amounts of aerosols in between 10 and 20 μm were detected (Price et al., 2018;Sanchez-Marroquin et al., 2019;Sanchez-Marroquin et al., 2020;Sanchez-Marroquin et al., 2021). Most of the detected particles were below ~2 μm. At sizes below ~3 μm, the comparison between the optical probes and the SEM-EDS size distributions are consistent in most cases, with an undercounting at the lower end of the SEM-EDS technique (~0.3 μm). This undercounting is related to the difficulty in observing small organic rich particles and has been discussed in Sanchez-Marroquin et al. (2019). At sizes above ~3 μm, the optical probes and SEM-EDS size distributions showed a comparable amount of detected particles in samples C089_3 and C090_1. However, for samples C087_1 and C091_2, the optical counters detected a larger concentration of particles with sizes ~5 to 10 μm than the SEM analysis of the filters. Similar discrepancies have been observed previously with these instruments in another low aerosol environment (Young et al., 2016), and were attributed to regions of high humidity even if the average humidity in a run should not have led to substantial hygroscopic growth. In dust plumes near Iceland and in aerosol around the UK where there was a significant coarse mode the agreement between CDP and SEM tended to be good. We note CDP is designed for cloud droplets, and we are using it at the edge of its capability for larger aerosol particles, hence there may be some biases which seem more significant in low aerosol environments.

In terms of chemical composition, the samples were mainly dominated by sea spray (Na rich) and mineral dust (Si rich, Si only, Al-Si rich and Ca rich) particles. There were some smaller contributions of S rich particles (likely sulphates) and Carbonaceous particles (likely black carbon or organic material). This is consistent with other SEM-EDS studies of the aerosol samples collected on the Alaskan Arctic from the ground (Chen et al., 2022;Creamean et al., 2018;Kirpes et al., 2018;Gunsch et al., 2017) or during a ship campaign (Kirpes et al., 2020). However, we tend to observe larger fractions

of dust aerosol particles, particularly in the sample C090_1, where this type of aerosol constituted ~65
% of the surface area of the sample.
In this dataset, nearly all particles in the Na rich category were dominated by the presence of Na and
Cl, having traces of other elements (such as S in some occasions), consistent with sea spray particles.
As a consequence, we will refer to particles in this category as sea spray aerosol particles. Some
carbonaceous particles were also detected through most sizes and there were significant contributions
of S rich aerosol, particularly in the accumulation mode. As shown in Fig. 4 and Table 2, the surface
area of samples C087_1 and C091_2 were dominated by sea spray aerosol particles with sizes around
~ 1 μm. In Sect. S1 it is shown that most of the air masses associated with these samples had been
circulating above the Arctic Ocean at relatively low altitude (below 1000 m) before sampling took place.
This is consistent with the fact that sea spray aerosol particles are normally emitted by bubble bursting
in the surface of the oceans (Lewis and Schwartz, 2004). It is possible that the detected sea spray aerosol
in our study was transported from ice free oceans. However, Sect. S1 indicates that the closest ocean
masses were almost fully covered by sea ice (with some open leads) during the campaign and the
majority of the sampled air masses did not pass by the open oceans prior to sampling. Hence, it is
possible that the sea spray particles had been emitted from open leads in the sea ice, as this is thought
to be a source of sea spray aerosol in the region (May et al., 2016;Kirpes et al., 2019;Chen et al., 2022).
It is also possible that some of the sea spray aerosol has been directly emitted from the sea ice through
blowing snow events (Yang et al., 2008;Huang and Jaeglé, 2017;Frey et al., 2020).
Particles in the categories Si rich, Si only, Al-Si rich and Ca rich have a chemical composition consistent
with mineral dust particles so we will refer to them collectively as mineral dust. However, it should be
borne in mind that the composition of particles in these categories is also consistent with some types of
combustion ashes or volcanic ash. Mineral dust particles were present in all the samples, particularly
with sizes between 1 and 5 μm, constituting a substantial percentage of its surface area, as shown in
Table 2. This was particularly the case in the sample C090_1, where 65 % of the surface area was given
by mineral dust particles. Although we cannot fully determine the relative contribution of different
sources to the detected mineral dust, several arguments suggest that the sampled mineral dust originated
from the low latitude deserts. The back trajectory analysis shown in Sect. S1 suggests that most of the
air masses had been circulating around the sampling location prior to sampling for ~5 days. However,
the majority of the potential high-latitude dust sources were covered by snow at this time, so it seems
unlikley that this mineral dust is related to natural emissions, although we cannot rule out sources
associated with human activities along the coast (e.g. Purdue Bay oil fields). Mineral dust originating
from the Sahara and Central Asia is known to be transported to the Arctic, especially in late winter and
early spring, when this study took place (VanCuren et al., 2012;Fan, 2013;Huang et al., 2015;Francis
et al., 2018;Shi et al., 2022;Zhao et al., 2022). This is consistent with the some of the backtrajectories
associated to the samples collected on the C085 flight, which originate from Asia. Almost all the mineral
dust particles found in this study had sizes below 5 μm and it is known that dust particles have a lifetime
of many days so can conceivably be transported to Alaska from distant sources (Huneeus et al.,
2011;Ménégoz et al., 2012). Once in the Arctic, accumulation mode aerosol has a lifetime extending to
months during winter and spring, when removal processes are weak (Carslaw, 2022). The small sizes
of dust particles found in this campaign contrast with results obtained using similar techniques on
samples collected closer to dust sources, where dust particles with sizes above 10 μm are frequent (Price
et al., 2018;Ryder et al., 2018;Sanchez-Marroquin et al., 2020). Although this evidence suggests that
most of our dust likely originated in arid lower-latitude deserts, high-latitude dust could still contribute
to the dust budget or even dominate it during other times of the year such as autumn (Groot Zwaaftink
et al., 2016).
As shown in Table 2, C087_1 and C091_2 samples have a larger surface area of sea spray aerosol
particles (Na rich) than mineral dust, whereas sample C090_1 is dominated by the presence of mineral
dust. Hence, it is reasonable to ask if the mineral dust or organic material associated with sea spray is

the more important INP type in these samples. To estimate the relative contribution of mineral dust and sea spray aerosol to the INP population, we present the expected INP concentrations based on the SEM-EDS surface areas in Fig. 5, in comparison with the measured INP concentrations. The INP concentrations expected from the SEM-EDS analysis were calculated assuming a dust containing 10 % of K-feldspar (Harrison et al., 2019) (the ice active component of desert dust) and the parametrization of fertile soils given by O'Sullivan et al. (2014). Note that the latter is very similar to the desert dust parameterization given by Ullrich et al. (2017). For the pristine sea spray INP, the parametrization given by McCluskey et al. (2018) that links INP concentration and aerosol surface area has been used. As shown, even in the cases where there is more sea spray aerosol than mineral dust (C087_1 and C091_2), the minimum contribution of mineral dust INP is orders of magnitude above the INPs produced by the pristine sea spray aerosol particles. It is possible that the sea spray in this location was more active than defined by McCluskey et al. (2018), however,  the INP concentrations calculated based on the presence of dusts better explains the observed INP concentrations measured using the droplet freezing assay at the lower end of the temperature spectrum. At the higher end of the temperature spectrum, the measured INP concentrations are above those expected from a 10 % K-feldspar dust, but are consistent with the fertile soil dust parameterisation. It is known that fertile soil dusts contain biological ice nucleating material (O'Sullivan et al., 2014), hence this suggests that the samples from Alaska contained some biological ice nucleating material (either from marine or terrestrial sources). Although our INP concentrations would also be comparable with those predicted using the desert dust parameterization by Ullrich et al. (2017), the latter is usually higher than the activity of samples of airborne desert dust at temperatures greater than about -20°C from other studies (Boose et al., 2016;Price et al., 2018;Harrison et al., 2022;Reicher et al., 2018;Gong et al., 2020). It has been suggested that dust that has been transported far from its source regions is less active than arid soil dusts that have been recently aerosolised and also there appears to be substantial differences in activity of dust from different source regions (Boose et al., 2016;Harrison et al., 2022). Hence, we suggest that the enhanced ice-nucleation ability of our samples is perhaps due to the presence of biological material. This is consistent with other studies that also provided evidence that Arctic INP samples have a substantial biological component (Wex et al., 2019;Creamean et al., 2019;Santl-Temkiv et al., 2019;Porter et al., 2022).

## 5.  Conclusions

In this study, we present a new dataset of INP and SEM-EDS aerosol size-resolved composition measurements in the Western North American Arctic in March 2018. Back trajectory analysis suggests that most of  these air masses spend the preceding five days circulating over or near Alaska and Northern Canada where local sources of primary aerosol were supressed by snow and ice cover. Observed INP concentrations were at or close to the limit of detection of the measuring technique, being always below 0.1 and 1 $L^{-1}$ at – 15 $^{\circ}$C and -20 $^{\circ}$C respectively. SEM-EDS analysis revealed that samples are mostly dominated by the presence of mineral dust and sea spray aerosol particles, with some contributions of sulphur rich and carbonaceous particles. The mineral dust is most likely sourced from the low-latitudes, rather than local high-latitude dust sources. Our analysis shows that mineral dust will always contribute more INP to the INP population than sea spray, despite sea spray being more abundant in some samples. However, it appears that the ice-active mineral K-Feldspar cannot account for all of the observed INPs, especially above ~ -22°C. This suggests that there is another INP type that controls the INP spectrum above -22°C; these particles may be biogenic in origin, but where this biogenic ice nucleating material might be derived from is unclear. More work is clearly required to understand the sources and nature of INP in the winter and early springtime Arctic.

**Acknowledgments**

We are grateful to all the people involved in the MACSSIMIZE campaign, led by Chawn Harlow (UK Met Office). The samples were collected using the FAAM BAe-146-301 Atmospheric Research Aircraft, flown by Airtask Ltd., maintained by Avalon Aero Ltd., and managed by FAAM Airborne Laboratory, jointly operated by UKRI and the University of Leeds. We acknowledge the Centre for Environmental Data Analysis for the access to the FAAM datasets used here. We would also like to thank Duncan Hedges and Richard Walshlaw at the Leeds Electron Microscopy and Spectroscopy Centre.

**Author contributions**

Aerosol measurements during the MACSSIMIZE campaign were organised by ASM, JBM, and BJM. ASM and BJM worked on the manuscript with contributions from all authors. The field work was carried out by ASM and JBM. ASM performed all the experimental measurements (INP analysis and SEM-EDS). The SEM-EDS technique was developed by ASM and ITB. The back-trajectory analysis was carried out by SLB and ASM. All the authors contributed to the discussion.

**Financial support**

This research has been supported by the European Research Council (MarineIce (grant no. 648661)) and the Natural Environment Research Council (NE/R006687/1, NE/T00648X/1).

**Competing interests**

The authors declare that they have no competing interests.

**Data and materials availability**

All data needed to evaluate the conclusions in the paper are present in the paper and/or the Supplementary Materials. The digitalized data are available from https://doi.org/xx.xxxx/xxx. FAAM data associated to the flights can be found in: Facility for Airborne Atmospheric Measurements; Natural Environment Research Council; Met Office (2018): FAAM C085 MACSSIMIZE flight: Airborne atmospheric measurements from core and non-core instrument suites on board the BAE-146 aircraft. Centre for Environmental Data Analysis. https://catalogue.ceda.ac.uk/uuid/b04281cc10c44d9dab1eb2e4eb19d5b8.

| Sample | Date (2018) | Start time (UTC) | End time (UTC) | GPS altitude (m) | Radar altitude (m) | BL or FT | Vol. PC (L) | Vol. tef. (L) | PTFE position | Stored | Temperature (ºC) | Dew temperature (ºC) | Aerosol concentration (cm-3) |
|---|---|---|---|---|---|---|---|---|---|---|---|---|---|
| C085_1 | 03/11th | 22:22 | 22:34 | 475 | 474 | FT | 466 | 312 | Up | No | -11.1 | -14.4 | - |
| C085_3 | 03/11th | 23:18 | 23:40 | 604 | 546 | FT | 461 | 355 | Low | No | -5.4 | -10.6 | - |
| C086_1 | 03/13th | 21:14 | 21:22 | 38 | 38 | BL | 212 | 159 | Low | No | -16.8 | - | 76.4 |
| C086_2 | 03/13th | 21:29 | 21:49 | 138 | 139 | BL | 231 | 143 | Up | No | -17.9 | -18.3 | 75.9 |
| C086_3 | 03/13th | 22:11 | 22:31 | 386 | 387 | Intersection | 644 | 209 | Low | No | -11.3 | -14.1 | 35.3 |
| C087_1 | 03/16th | 20:44 | 21:26 | 310 | 309 | BL | 1047 | 565 | Low | No | -19.7 | - | 68.9 |
| C087_2 | 03/16th | 21:33 | 22:03 | 304 | 305 | BL | 965 | 447 | Up | No | -16.4 | - | 61.2 |
| C087_3 | 03/16th | 22:30 | 22:44 | 536 | 491 | FT | 392 | 217 | Low | Yes | -13.6 | - | 46.8 |
| C089_1 | 03/18th | 18:01 | 18:42 | 584 | 522 | FT * | 1198 | 714 | Low | No | -21.3 | -20.2 | 40.3 |
| C089_2 | 03/18th | 18:49 | 19:17 | 573 | 506 | FT * | - | 398 | Low | No | -21.2 | -19.3 | 45.2 |
| C089_3 | 03/18th | 19:28 | 19:48 | 591 | 557 | FT * | 404 | 214 | Up | Yes | -20.9 | -18.8 | 47.7 |
| C090_1 | 03/20th | 20:15 | 20:38 | 547 | 487 | FT * | 735 | 349 | Low | No | -15 | -15.4 | 62.1 |
| C090_2 | 03/20th | 20:53 | 21:26 | 563 | 503 | FT * | 488 | 409 | Up | No | -14.6 | -15.6 | 62.3 |
| C091_2 | 03/21th | 18:27 | 18:56 | 122 | 123 | FT * | 1187 | 376 | Up | No | -28 | - | 63.6 |
| C091_3 | 03/21th | 19:01 | 19:14 | 295 | 297 | FT * | 644 | 203 | Low | Yes | -25.7 | - | 63.9 |
| C091_4 | 03/21th | 19:21 | 19:51 | 71 | 68 | FT * | 635 | 635 | Up | No | -29.8 | -27.4 | 31.8 |

Table 1. Details of the samples collected during the MACSSIMIZE campaign. PTFE position
refers to which inlet was used to collect the PTFE sample in each run. The other line was used
to collect the polycarbonate sample. In order to determine if the sample was collected within
the boundary layer (BL) or in the Free Troposphere (FT), we looked at the temperature and
potential temperature profiles. (*) For all the runs in the C089, C090 and C091, the flight did
not descend low enough to determine the exact depth of the BL. Hence, it was assumed that
the runs occurred above the BL. Stored filters were kept for a few hours or days at -18 ºC,
while the rest of them were analysed immediately after collection without any long-term
storage. The given altitude values correspond to the average of each run. The mean values of
the air temperature across the run was derived from the Rosemount de-iced temperature
sensor, while the dew temperature is given by the Buck CR2 Hygrometer of the BAe-146.
Dew temperature could not be calculated for all runs due to technical problems. The aerosol
number concentration corresponds to the range of ~0.1 to ~3 μm and it has been calculated
using the PCASP instrument. Blank entries correspond to a filter that was not collected or the
instruments not working.

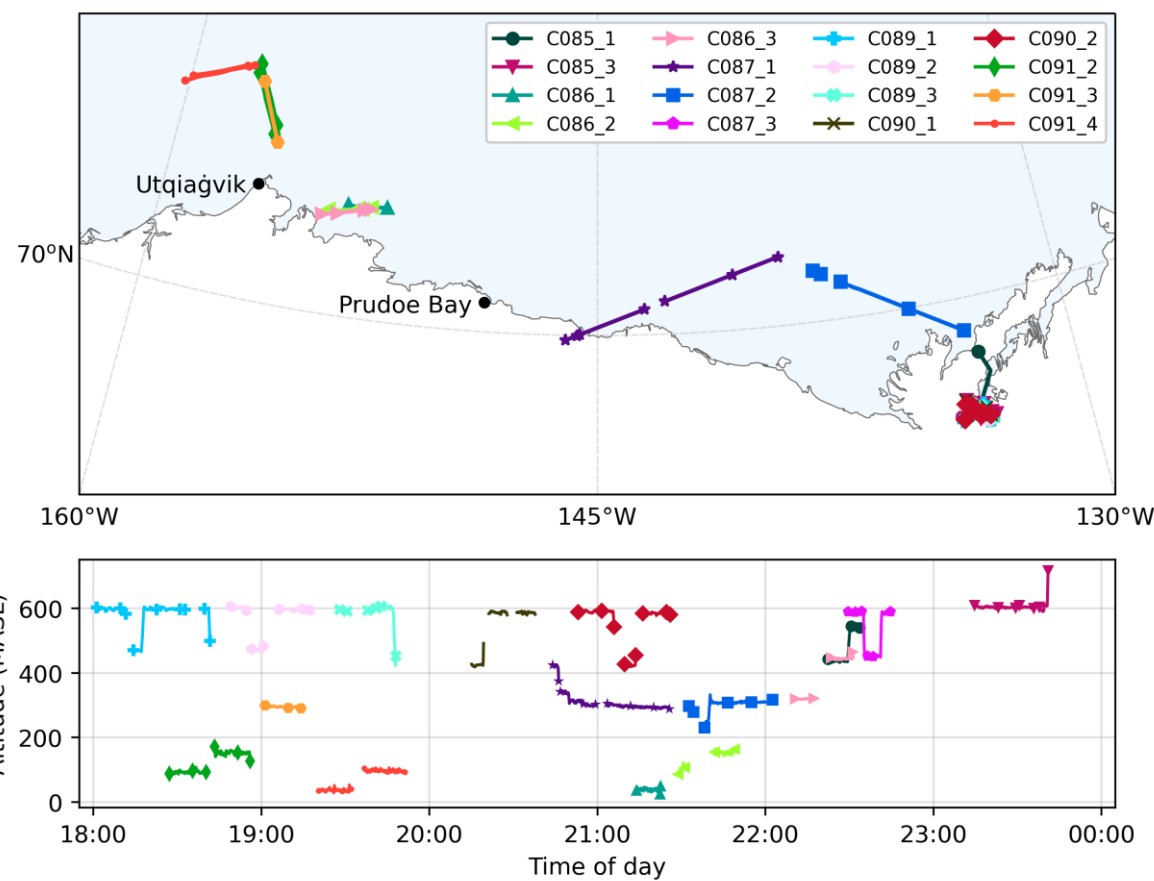


Figure 1. Flight tracks of the samples collected in this study and described in Table 1 (top panel). GPS altitude at which the samples were collected (lower panel). The altitude is presented against the GTM time at which the samples were collected (although they were collected across several days).


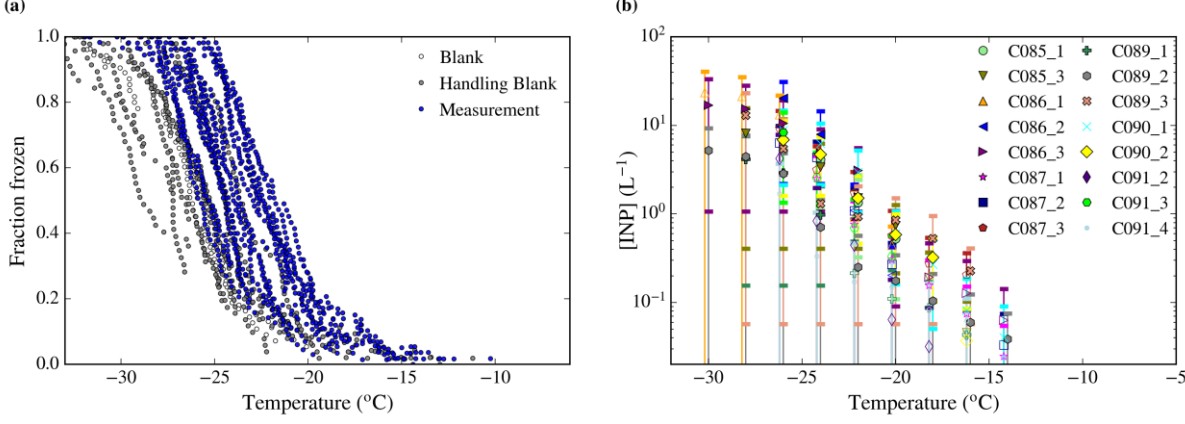


Figure 2. (a) Fraction of droplets frozen for all filter samples as well as blanks and handling blanks. (b) INP particle concentrations for each filter sample. Data points corresponding to the upper limits (open symbols) have been shifted 0.2 ºC along the x-axis for clarity. The way in which the INP concentrations, upper limits and its uncertainties have been calculated are shown in Appendix A. The criteria to determine if a measurement is above the limit of detection is based on 68% confidence intervals.

453

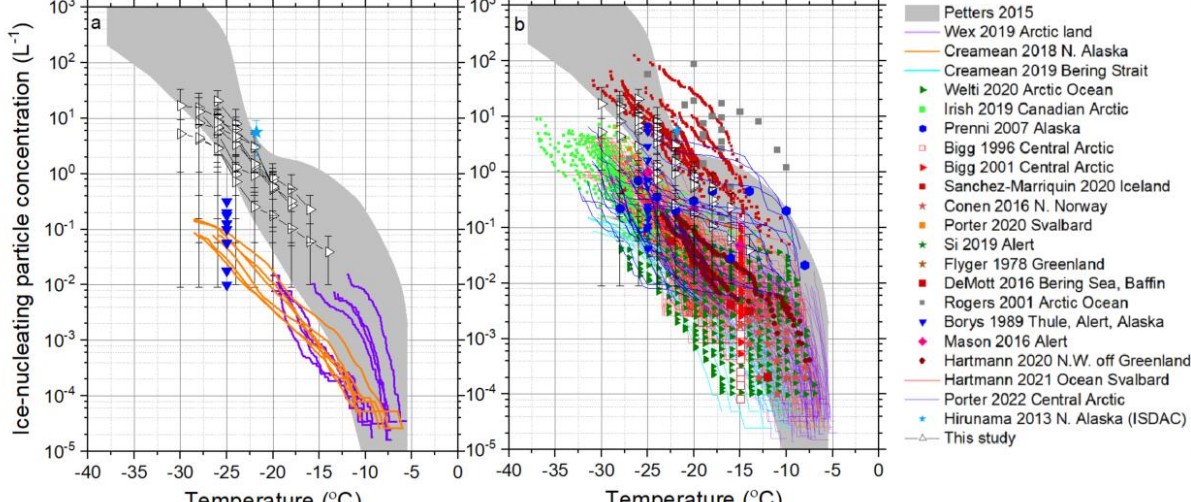

Figure 3. INP concentrations from the present study compared with literature data. We only show our data that was above the background (limiting values are included in Figure 2). Note that this data is above the limit of detection based on 68% confidence intervals. The left panel is limited to a comparison with previous measurements at nearby locations at a similar time of year (February, March and April) (Borys, 1989;Creamean et al., 2018;Wex et al., 2019;Hiranuma et al., 2013). We also limit this comparison to data recorded at or above water saturation, which limits the data from Hiranuma et al. (2013) to a single point during what they describe as a relatively high INP concentration event. Note that for the dataset of Wex et al. (2019), the concentrations increased through this period with the two highest INP spectra from April. The right hand figure is a comparison with Arctic data in general, from any time of the year and any location (Flyger and Heidam, 1978;Borys, 1989;Bigg, 1996;Rogers et al., 2001;Bigg and Leck, 2001;Prenni et al., 2007;Hiranuma et al., 2013;Conen et al., 2016;DeMott et al., 2016;Mason et al., 2016;Creamean et al., 2018;Wex et al., 2019;Creamean et al., 2019;Irish et al., 2019;Si et al., 2019;Porter et al., 2020;Sanchez-Marroquin et al., 2020;Welti et al., 2020;Hartmann et al., 2020;Hartmann et al., 2021;Porter et al., 2022). The mid-latitude data range given by Petters and Wright (2015) is also shown.

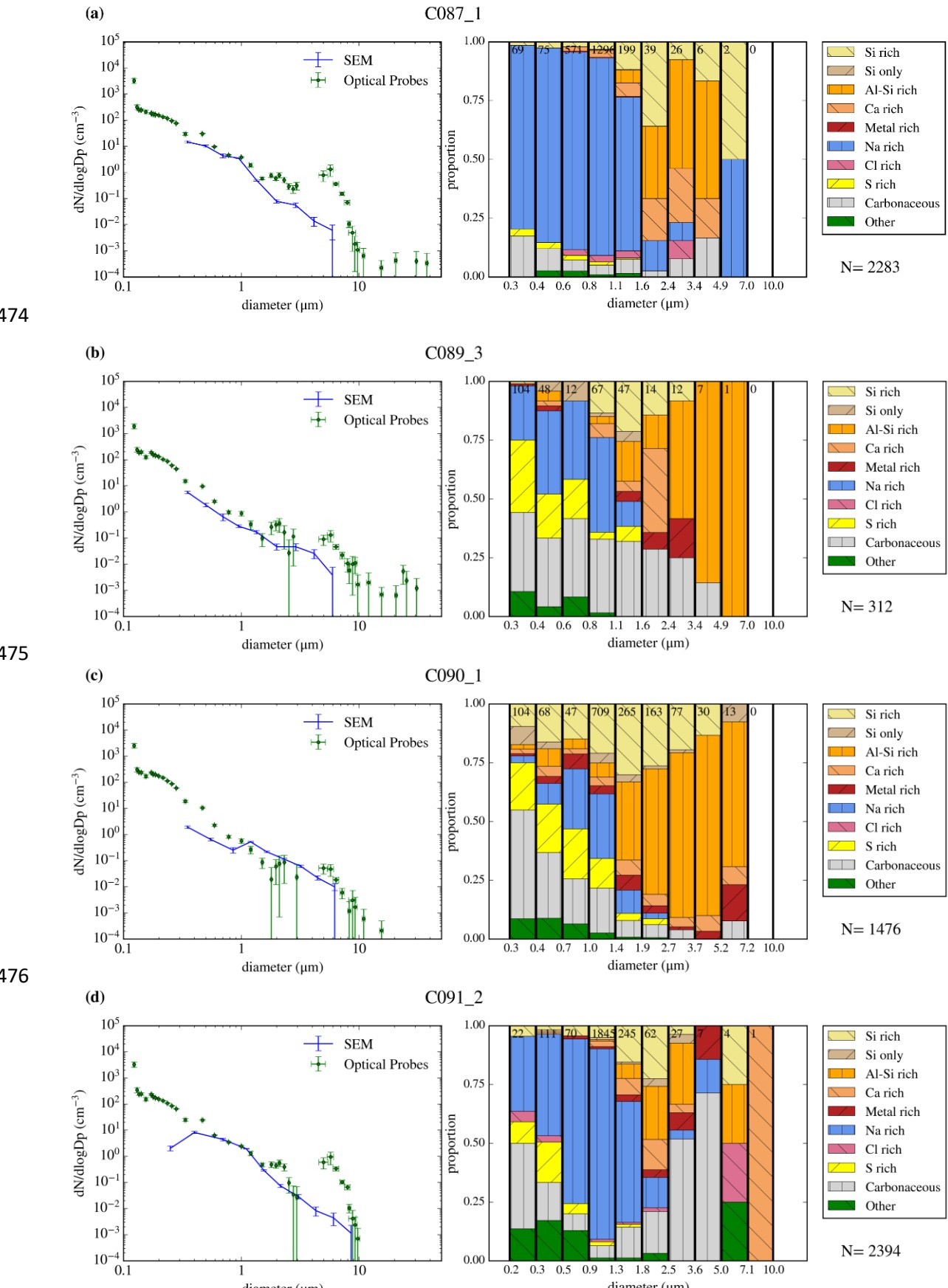





Figure 4. Results of SEM-EDS analysis of each analysed sample (a-d) showing comparison between SEM-EDS and PCASP-CDP number size distribution (left) alongside number size-resolved composition fractions (right).

| Sample | Dust area (µm²/cm³) | Dust limit of detection (µm²/cm³) | Dust area percentage | Sea spray aerosol area (µm²/cm³) | Sea spray area percentage |
|--------|---------------------|-----------------------------------|----------------------|----------------------------------|---------------------------|
| C087_1 | 0.75 | 0.042 | 13.9 | 3.97 | 73.4 |
| C089_3 | 0.57 | 0.15 | 38.1 | 0.26 | 17.1 |
| C090_1 | 1.21 | 0.083 | 65.5 | 0.16 | 8.9 |
| C091_2 | 0.53 | 0.051 | 11.3 | 2.79 | 59.5 |

Table 2. Surface area of dust and sea spray aerosol from SEM-EDS analysis. The dust limit of detection corresponds to the upper limit of the dust concentration detected on the handling blank filter based on one standard deviation. Note that the given dust and sea spray aerosol percentages refer to surface area percentages. The limit of detection of sea spray aerosol particles has not been indicated because the presence of this type of particles in the handling blank is negligible. Further information on the size-resolved composition of the handling blanks and a discussion about it can be found in (Sanchez-Marroquin et al., 2019).

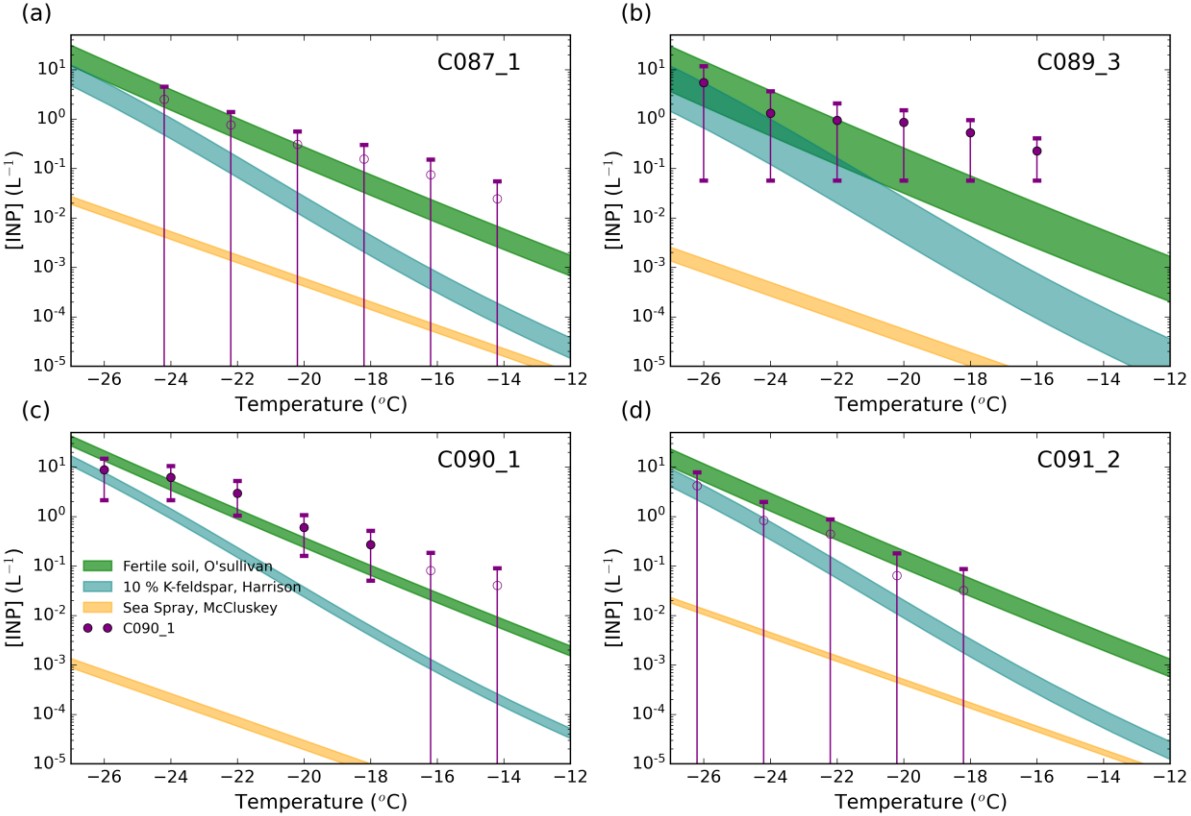

Figure 5. Predicted INP concentration of the SEM-EDS samples compared with the INP measurements at -20 °C. The dust INP prediction has been calculated by applying different ice-nucleation parametrizations to the surface area of dust calculated from the SEM-EDS analysis. The O'Sullivan et al. (2014) for fertile soils and a dust containing 10 % of K-Feldspar (Harrison et al., 2019) have been used. The NaCl INP prediction has been obtained by applying the sea spray aerosol parametrization

from McCluskey et al. (2018) to the SEM-EDS sea spray aerosol surface area. The purple points
correspond to our INP measurements or upper limits, based on 68% confidence intervals (Appendix A).

**Appendix A: Upper limit determination and background subtraction of the ice-nucleation**
**experiments**

As shown in Fig. 2a, most of the fraction of droplets frozen produced by the collected samples were
comparable or only slightly above to the ones produced by the handling blanks. Hence, we established
criteria to separate data points of the INP spectrum that are not significantly above the limit of detection
of the instrument. The analysis is performed using the differential spectrum of ice-nucleus rather than
the cumulative spectrum, which is normally used to display and compare ice-nucleation data such as
INP concentrations and densities of active sites (Vali, 1971;Vali, 2019). First, we create a histogram
with the number of freezing events per temperature interval per sample. This is done for all the samples
and handling blanks, with temperature intervals of 2 $^{\circ}$C. We transform the number of freezing events
per interval of each sample into the differential INP spectrum, $k(T)$, using Eq. 1 (Vali, 2019).

$$k(T) = -\frac{1}{V \Delta T} ln \left(1 - \frac{\Delta N}{N(T)}\right)$$

Eq. 1

In Eq. 1, $V$ is the droplet volume, $\Delta T$ is the temperature interval, $\Delta N$ is the number of frozen droplets
between $T$ and $(T$-$\Delta T)$, and $N(T)$ is the number of unfrozen droplets at $T$. The $k(T)$ values of the handling
blanks is shown in Fig B1, alongside the mean value of each interval and its standard deviation. Note
that many of the temperature intervals had zero freezing events, corresponding to $k$ equal to zero. These
zero values cannot be seen in Fig. B1 but they have been included in the means and standard deviations.
The mean and standard deviation of the $k$ values produced by each handling blank has been compared
with the k values corresponding to each sample. The uncertainty in the $k$ values associated with each
sample has been calculated using a very similar Monte Carlo simulation as used previously (Vali, 2019)
using a 68 % interval. The $k$ values associated to each sample were individually compared with the
mean and standard deviation of the $k$ values of the handling blanks. A data point was considered above
the limit of detection when its lower error yields above the mean plus standard deviation of the blanks.
Background subtraction was applied to data points significantly above the limit of detection. This was
done by subtracting the mean of the $k$ values of the handling blanks. The error of the background-
subtracted point was calculated by square rooting the quadratic sum of the error of the $k_{sample}$ and
$k_{background}$. Two examples of the comparisons between samples and the handling blanks are shown in
Fig. B2. (a) corresponds to a case where no data point was higher than the limit of detection, while (b)
corresponds to a case where most of the data points were significantly above the limit of detection. Note
that all the data measured on the 16[th] of March (flight C087) has been flagged as an upper limit. This is
because the handling blank experiment carried out on that day was unusually high, being compatible
with all the measurements.
The background corrected $k(T)$ was integrated into the cumulative spectrum of active sites, $K(T)$, using
Eq. 2 (Vali, 1971;Vali, 2019).

$$K(T) = \sum_{T=0}^{T} k(T) \Delta T$$

Eq. 2

INP concentrations were calculated from , $K(T)$ using Eq. 3, where $V_d$ is the droplet volume, $A_{filt}$ is the
area of the filter, $V_a$ is the sampled air volume and $\alpha$ is the contact surface of the droplets. For this study,
we used the same values than Sanchez-Marroquin et al. (2021).

$$INP(T) = \frac{K(T)V_D A_{fil}}{V_a \alpha}$$

Eq. 3

A $k$ value which was not significantly above the limit of detection has been represented with lower bars
going to zero in the INP spectrum (meaning upper limit to the INP concentration). However, if a $k$ value
not significantly above the limit of detection was preceded by a value which was above the limit of
detection, then as a result of the cumulative nature of the reported INP concentration the corresponding
value is reported with a filled symbol, but the lower bound of the error bar does not change since it is
possible that no new INP were present in that temperature interval. In Fig B3 one can see the INP
concentration of all the samples collected in this study per each day.

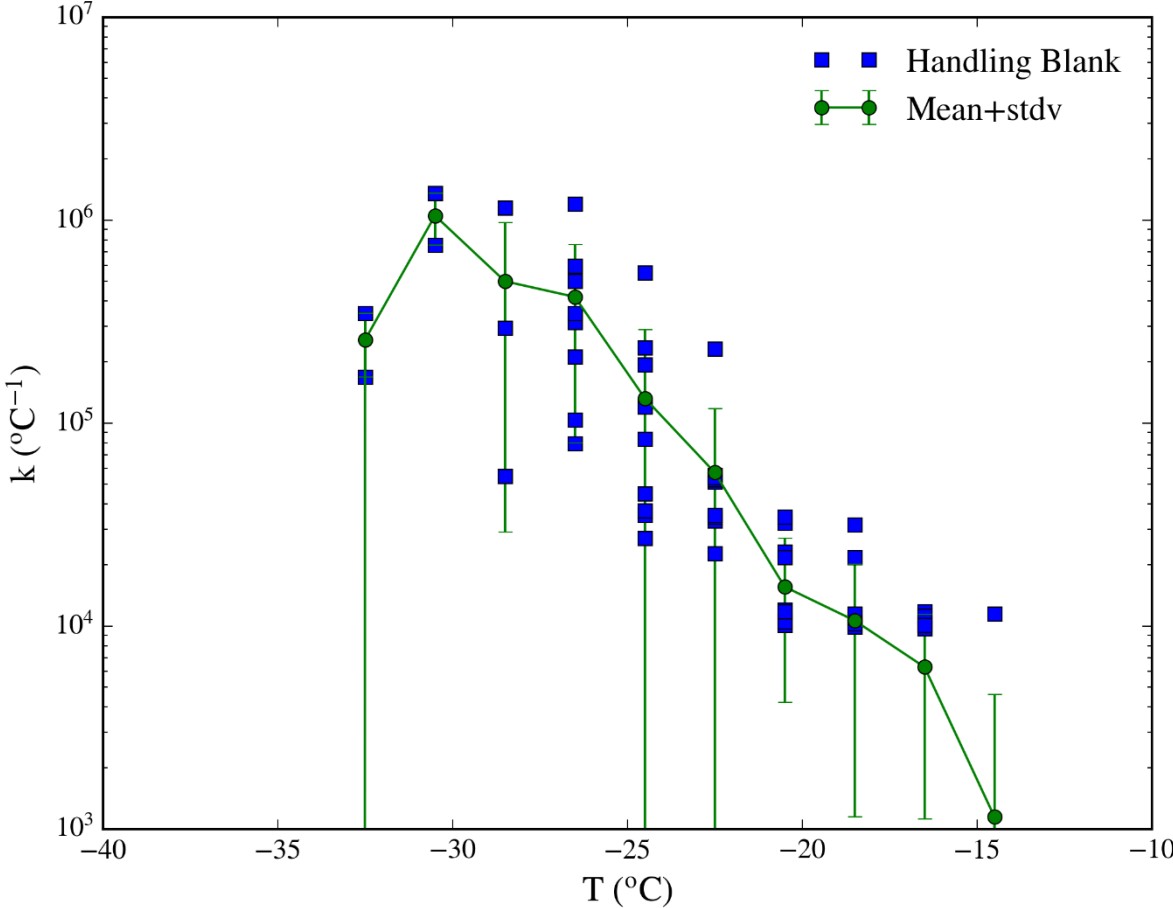


Figure B1. Differential spectrum of ice-nucleus of all the handling blanks performed during this
campaign. Data is shown in blue, while the mean and standard deviation of the data of each bin are
show in green.

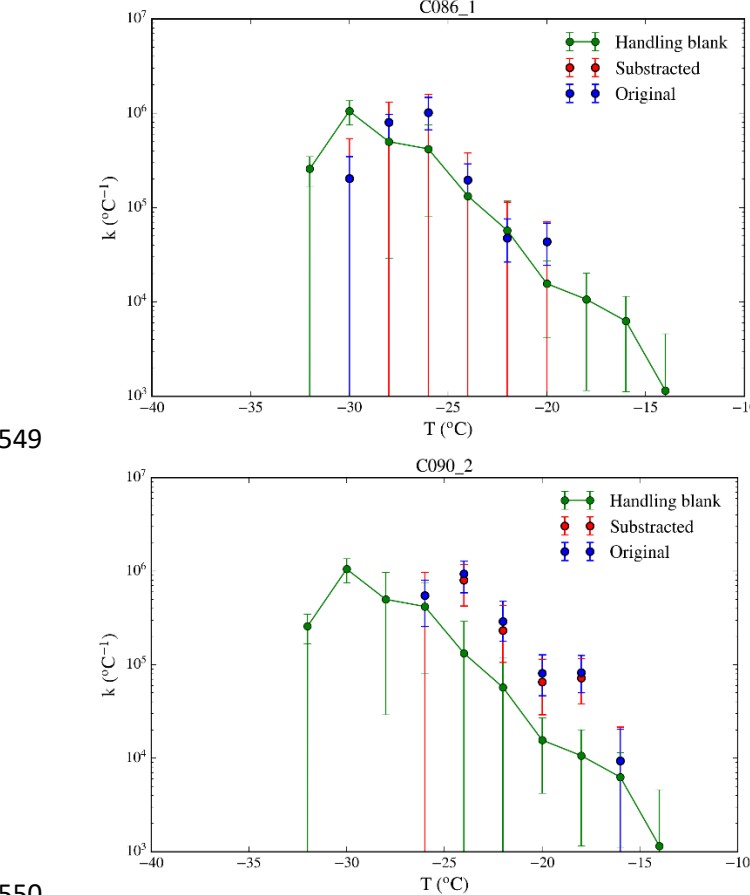



Figure B2. Examples of a comparison between the handling blank mean with two samples. None of the data points of sample C086_1 is significantly above the background. However, most of the data points associated with sample C090_2 are more than one error bar above the data produced by the handling blanks and they have been background-subtracted.

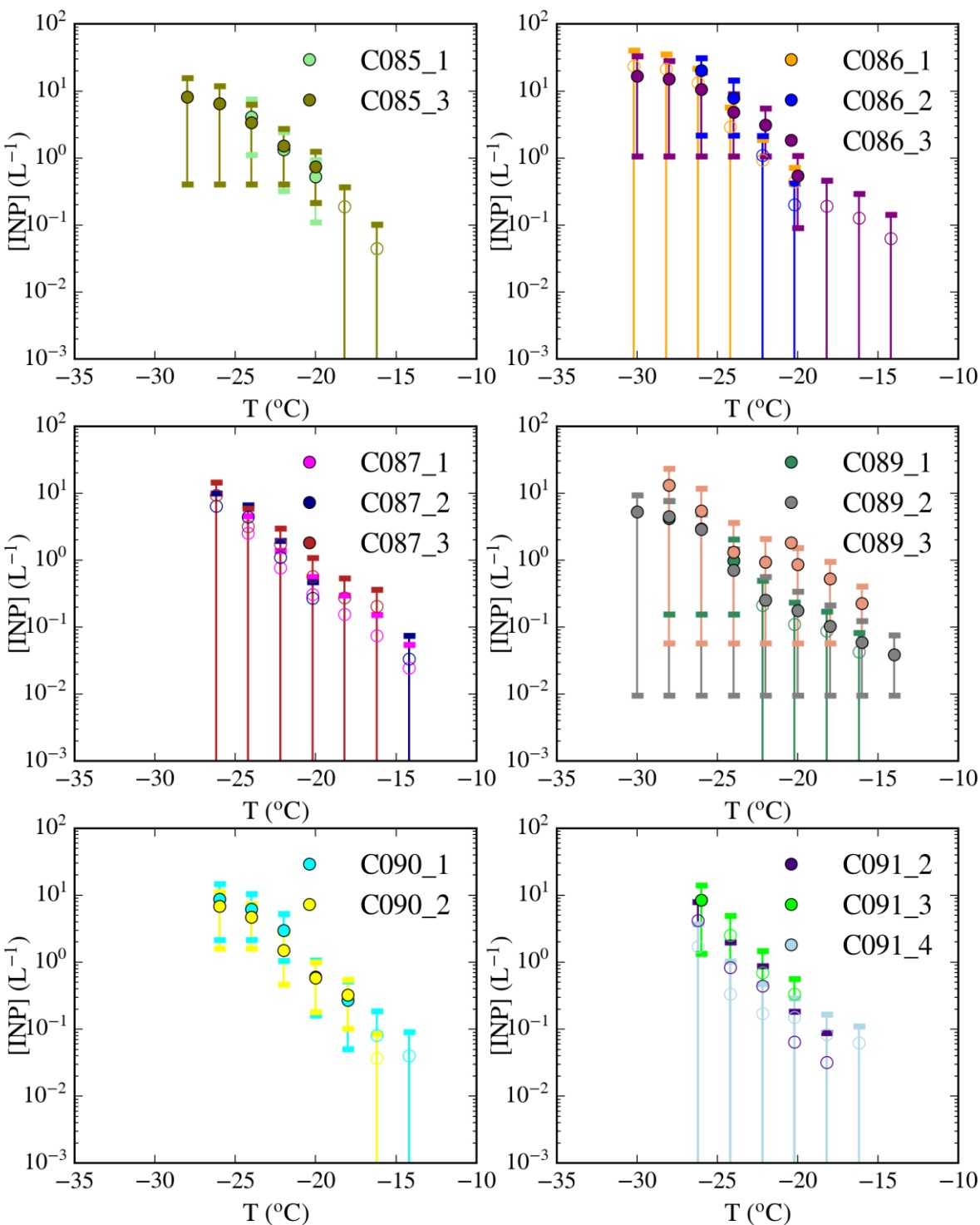

Fig B3. INP concentrations and upper limits shown in Fig. 2 separated per sampling day. A list of the
days when these samples were collected is shown in Table 1. Note that full markers corresponds to
measurements above the limit of detection, while hollow markers correspond to upper limits. This has
not been specified in the legend as some samples have both upper limits and measurements at the same
time.

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
