# Peer review of "Aircraft ice-nucleating particle and aerosol composition measurements in the Western North American Arctic"

_Atmospheric Chemistry and Physics, 2022_

## Referee Comment (RC1)

Sanchez-Marroquin et al. describe findings from an aircraft campaign that took place in mid-March 2018 primarily on the north coast of Alaska and the adjacent Arctic Ocean. They collected filter samples of the encountered aerosol and analyzed them with a freezing assay to derive the number concentrations of ice-nucleating particles (INPs), and also used SEM-EDS to obtain a size-resolved composition of the aerosol particles. In addition, they examined back trajectories and compared their results with previous studies and parameterizations. The measured INP spectra were always close to the background and all samples very little variation. They conclude that their samples are dominated by mineral dust from low-latitude sources. They also attribute the ice activity of the samples (below -22°C) to mineral dust and speculate biogenic INPs control the INP spectrum above -22°C. The study lacks exciting results, but that should not be held against the authors. The data pool on Arctic INP is limited, especially for measurements on aircraft. Therefore, any addition to this pool is valuable in itself because it helps us as the authors themselves write, "[...] understand the sources and nature of INP in the Arctic during winter and early spring". Overall, I recommend this manuscript for publication in ACP after some of its shortcomings have been addressed (see general and specific comments below)

General comments

1. A bit more discussion of aerosol-cloud interaction is needed. I.e. what do the results mean for mixed phased clouds and the overall INP "budget" in this region?
2. The authors suspect low-latitude sources for the mineral dust, is it possible to be more specific in that regard? It is known that mineral dust sources have compositional fingerprints (Scheuvens & Kandler 2014, and references therein, https://doi.org/10.1007/978-94-017-8978-3_2). Could the authors use the SEM-EDS results to constrain possible source regions? Could the authors also use satellite products to at least qualitatively verify the transport of dust from low to high latitudes over the relevant time period?
3. Several details on the flights and the sampling strategy are missing in my view, but are needed for a reader to assess the conditions in which the samples were collected. Hence I suggest the addition of the following information to the manuscript (at least in the appendix)
   - Individual Flight tracks and/or height profiles with the sections when a samples was collected highlighted.
   - Indicating if a samples was collected within the PBL/MBL or in the free troposhere; Indicating if sample was collected below/above a cloud layer. This could be done as additional columns in Table A1
   - If available, vertical profiles of aerosol number concentration (e.g. derived by integrating the OPC size distribution) and basic meteorological parameters (temperature, humidity)
4. For details on the inlet, the authors refer to one of their previous publications, which is fine, but nevertheless the main key should be repeated in this manuscript. That includes cut-off diameter, is the inlet heated or is the aerosol stream dried in some way to avoid the collection of cloud droplets, is the sampling isokinetic, and where on the aircraft is the inlet located?
5. Analogous to my previous comment, the main key points of the SEM-EDS should be repeated in this manuscript and not just referenced. This includes whether there was any form of sample preparation, such as gold sputter coating, what accelerating voltage was used, how long was the collection time for an EDS spectrum, whether the analysis was manual or automatic, (if the

latter, whether particle detection was based on contrast in the BSE image or something else, and whether possible artifacts were removed before further analysis).

L64
Since Tobo et al. 2019 has already been cited above, and because it is consistent with one of your conclusions, it should be explicitly mentioned that dust can also be a carrier for biological INPs

L72-73
This statement about Rinaldi et al.in relation to Wex et al. is correct, but also a bit misleading. Wex et al. attribute the seasonal increase to biological INP and see their effect primarily at high temperatures, with a less pronounced seasonality at lower temperatures. Because of the setup used by Rinaldi et al. they can measure at lower temperatures/higher concentrations. So even if there were a seasonality, Rinaldi et al. would not see it as pronounced in their data, as Wex et al. do. The wording here should take this into account.

L116-117
What is meant by "subset"? Were not all PC filters used for SEM-EDS? If so, what happened to the rest and how was this "subset" selected?

L153-155
I would suggest also including a more recent publication on aircraft INP measurements in the Arctic such as Hartmann et al. (2020), which you cited earlier, in the comparison. They differ in location but are very similar in time (March/April 2018). Borys (1989) may be more comparable in location, but his measurements took place more than 30 years ago, before the extraordinary warming of the Arctic was observed (Arctic Amplification), and some might argue that the differences are related to this.

L155
When making the comparison here, it should also be mentioned which method was used in the respective studies to determine [INP] (freezing assay, CFDC, expansion chamber).

L172
There are many ways to obtain particle diameters in SEM images, it should be mentioned what diameter was used here

L173
More details about the optical probes are needed, such as the name of the instrument, the manufacturer, and where it is mounted. Before reading in the caption of Fig. 4 that it was a PCASP-CDP, I wondered if the OPC might have been fed through the same inlet as the filter sampler. PCASP-CDP, on the other hand, indicates that it is a wing-mounted device.
In addition, it should be mentioned whether the OPC size distributions shown in Fig. 4 represent the average over the entire period in which the SEM sample was taken or whether it is one point measurement during that sampling period.

L175 (+ Fig 4)
Is there a reason why only four samples were analyzed with SEM-EDX? An explanation should be given in the text.

L190-192
PCASP-CDP suggests the combination of aerosol and cloud probe. Doesn't this make it possible to confirm rather than speculate whether the artifacts are actually cloud droplets?

L201
Because of the scale of the y-axis in Fig. 1b, it is difficult to tell when the air masses are below 500m as described in the text. A horizontal line indicating the 500m mark might help, and perhaps a logarithmic axis.

L204-207 (+ Fig 1a)
Looking at sea ice concentration (e.g. https://seaice.uni-bremen.de/databrowser/#day=11&month=2&year=2018&img=%7B"image"%3A"image-1"%2C"product"%3A"AMSR"%2C"type"%3A"nic"%2C"region"%3A"Arctic3125"%7D ) instead of sea ice extent or satellite imagery in general ( e.g., https://go.nasa.gov/3TevAJn ), it is clear that the ocean around the north coast of Alaska is not completely covered by ice, but is highly fragmented with numerous open leads. This makes the marine particle source mentioned in L207 more relevant in my eyes and should be discussed accordingly in the text.

L210-211
Defining mineral as described here is reasonable in many cases. However, considering that there are algae with Si or Ca-based skeletons and the question of the importance of marine or terrestrial INP sources is still debated, I wonder if algal skeletal fragments were misclassified as mineral dust. Can the authors comment on this? Was visual screening done to determine the amount of non-dusty Si and Ca containing particles?

L219-222
Indeed, studies such as Huang et al. (2015) show that dust from lower latitudes can reach the Arctic within 5 days. The back trajectory analysis in this study also states that the air masses containing the dust particles circulated for at least 5 days. Meaning that the found mineral dust particles with a size of 1-5 μm had to remain suspended in the air for a minimum of 10 days. Can the authors support the claim this typically happens in the atmosphere?

L234
Fig. 5 (instead of Fig. 3)?

L235-236
Is it not possible to use the SEM-EDS analysis to derive a K-feldspar amount that is more representative of the samples of this study? You already have the "Al-Si rich" category, which should contain mainly the feldspars and their weathering products (illite, kaolinite, etc.). If this category is filtered for appropriate amounts of K, an estimate for the K-feldspar content could be derived. I think studies by K. Kandler and co-authors have covered the categorization of mineral dust with SEM-EDS in detail.

L238-242
It should be noted that the parameterization presented by McClusky et al. (2018) is meant to represent

pristine SSA and they intentionally excluded events with elevated INP concentrations.  They also state t during a period of elevated marine organic aerosol from offshore biological activity, [INP] at -15°C was $7.7 \times 10^{-3}$ $L^{-1}$. A value that is above the mineral dust and below the fertile soil in Fig. 5.

This, of course, does not refute the author's conclusion that ice activity is related to mineral dust, but the possibility of biological INP of marine origin should be discussed.

McClusky et al. (2018) also present a different parameterization based on TOC. Perhaps the authors can derive an estimate for TOC based on the "carbonaceous" category of the SEM-EDS data and add these predicted INP concentrations to Fig. 5 as well.

L252-254
Following the author's train of thought, the sampled aerosol was transported for about 10 days, and as noted here, a decrease in ice activity would have been expected, yet Fig. 5 shows a very good match with fertile soils collected directly from the ground and brought to the laboratory quickly. This would suggest that the aerosol was not transported far, but may have more local sources. Can the authors comment on that?

In this context, could the use of monthly average snowpack (Fig. 1a) or the sensitivity of the satellite product used for the ERA5 reanalysis be obscuring exposed soils?

L255-256
In the INP community, heat and $H_2O_2$ treatments are commonly used to gain additional insight into the presence of proteinaceous and organic INPs. Could the authors perform such treatments as the results could further support the claim presented here?

Table A1, second column
2018 instead of 2017

Table A1
Are the presented altitude values averages over the sampling time? If yes, that should be mentioned in the text and caption.

Fig. 1 a)
A reader is not able to recognize where a trajectory starts/ends. If the daily marker would shrink with time, the reader might be able to identify the direction of the trajectory and hence start and end.

The authors also chose to show only one return trajectory per sample. For short sample times, a single back trajectory could be representative, but for longer sample times, the aircraft could travel long distances and enter air masses with very different histories. Can the authors comment on the representativeness of back trajectories they show. The authors might also consider adding a figure to the appendix showing high frequency back trajectories for each sample.

---

## Referee Comment (RC2)

This manuscript presents aircraft measurements of immersion mode ice-nucleating particles (INPs) in the Western North American Arctic in March. The size-resolved chemical composition is used to identify the sources of INP. The conclusion is not new, but the aircraft measurements of INP in the Arctic are valuable. The following major comments must be satisfactorily addressed before consideration for publication.

General comments:

1. Better to give more detailed information about the flight track and sampling strategy. For example, Appendix A shows the sampling time varies from ~10 to 30 mins. During the sampling, does the flight sample cloud or not? The flight height varies from 100 to 700 m, how much percent of the time is within the boundary layer of each sample?

2. The March in the Arctic is very often influenced by Arctic haze (not only long-range transport of dust but also anthropogenic pollution or biomass burning). Do you see any indication of anthropogenic pollution? For example, sample C089_3 shows a high carbonaceous fraction, what is the source of carbonaceous? Why only reported the chemical composition of four samples?

Minor comments:

1. Lines 67-79: It is better to include the recently published paper from Creamean et al. (2022), which discussed the seasonal variation of INP in the central Arctic.

2. Line 98: Add "open leads" after snow. Open leads are most likely omnipresent in the central Arctic. And later when you discuss the aerosol sources in Lines 201 -209, the sea salt aerosols are also likely from open leads.

3. Line 137: Change "approach to (Vali, 2019)" to "approach to Vali, (2019)".

4. Lines 190-192: Do the aircraft measurements also have cloud-related measurements, like liquid water content or cloud droplet number concentration?

5. Figure 4 and Table 1: I understand that Figure 4 left is the number size distribution. I assume Figure 4 right is the mass fraction of different components. My question is how to calculate the surface area of sea salt and dust in Table 1.

6. Line 365 Appendix B: Did you use the mean background value of all handling blank samples and subtract this value? The frozen fraction of handling blank samples shows a larger variation (Figure 2a). How to explain the larger range of backgrounds?

7. Figure B3: Please use the correct legends (solid or hollow) for each sample. For example, legend markers of C091_4 should be hollow.

**References:**

Creamean J M, Barry K, Hill T C J, et al. Annual cycle observations of aerosols capable of ice formation in central Arctic clouds[J]. Nature communications, 2022, 13(1): 1-12.

---

## Referee Comment (RC3)

Sanchez-Marroquin et al. present airborne measurements of INP concentrations and size-resolved, single-particle aerosol composition over coastal regions of the Pacific Arctic sector in March 2018. Main conclusions include aerosol composition dominated by sea spray and mineral dust. Further, the authors concluded that long-range transported fertile soil dust was likely the main contributor to the INP population during four cases. Reporting these measurements is important, given the limited observations of INPs in the Arctic, and especially observations above the ground. However, the manuscript has several major issues that should be addressed prior to publication in ACP.

**General comments:**

The introduction is too short and does not include enough relevant background. For instance, there is only a very brief introduction to Arctic mixed-phase clouds here. Some more effort should go into describing AMPCs, INP observations, and how limited they are relative to other latitudes. For the INP aspect, since this is an Arctic manuscript, the introduction material on INPs should focus more on what has been observed in that region. There are some missing key references and studies of Arctic INPs. See my specific comments below on these and what should be included. In general, *some* of the relevant literature is included, but should be more complete and involve discerning where and when the observations were made, since those can vary quite a bit depending on location, sea ice extent, etc. This is hinted at on lines 150-151, but should be elaborated on in the introduction to develop the broader picture for the reported observations.

Even though some of the methodologies are described elsewhere, pertinent information should at least be distilled here so the readers do not have to go on a wild goose chase to obtain such information. Much of it is buried in the appendix, but why not include it in the main text? Here are some questions I had after reading through the methods, information that could be briefly added for clarity:

- When I first read it, it was not clear if only 1 filter was collected per flight, for how long, which altitudes, etc. Table A1 has this information, which I think would be important to include in the main paper.
- There is not enough information on filter collection, preservation, and analysis in the main text. Again, this information is in Table A1 and its caption. Should be moved to the main text. For the altitude, was the aircraft spriling at that one altitude per filter? If so, what was the spatial coverage of these spirals? Or were these profiles and the one altitude value provided is an average? Need more details here.
- I originally assumed the processing was not done on site, however processing was indeed done on site as stated in the caption for Table A1. This may seem like mundane information, but how the filters are preserved and handled can have a large impact on their results, so is important enough to include in the main text.
- What filter pore size was used for the filters? Different filters have different collection transmission efficiency curves.
- Why was Teflon used for INPs? Teflon tends to have high backgrounds compared to polycarbonate. And why polycarbonate for microscopy? How were carbonaceous particles classified if the filter substrate material is carbon-rich?
- How exactly were the sample suspensions prepared? Shaken in water? For how long? What type of water was used? And was a blank water spectrum tested as well to see how much of the background is from the water itself versus the filter handling?
- How many drops and how many separate tests for the drop freezing experiments?

- What type of cold stage is this? Since it is not a commercial instrument, there needs to be some description of the method, even if just a few sentences.
- What is the temperature range of the cold stage?
- Was freezing detection manual (by human eye) or automatic?
- Why was a 68% confidence interval chosen? That seems like an oddly low one…
- I realize the inlet is described elsewhere, but clear referencing is needed, given aircraft inlets can be riddled with artifact issues.
- For HYSPLIT, why were the heights and 5-days chosen? A topic of discussion in the community is how long back is long enough. I don't have a good answer to that, but the authors should at least justify why they chose the parameters that were used. Additionally, typically a good approach is to run trajectory ensembles (i.e., every so many hours during filter collection and at multiple heights around the sampling point). One trajectory per sample is not enough, given the uncertainties with HYSPLIT, especially near the poles.
- There is hardly any information on the microscopy work. Even though it is in a previous paper, a few more sentences on the methodology should be included here. Where was the SEM-EDX analysis done? What about the details of how the samples were processed on the same filters? Were the samples frozen or stored at room temp and why? There should also be a brief synopsis of how the classifications were defined.

These are just a handful of questions I had, but effectively, the authors should take care to include the relevant methodology information in a succinct manner in the main text instead of in the appendix. That would alleviate most of my concerns here. The details on the calculations and blank corrections could remain in the appendix, if the authors desire, but the rest would be more helpful in the main text.

I am concerned about the background INP spectra. These are very high, and I suspect part of it is due to the use of Teflon and the fact that only a few hundred liters of air were collected per filter. Sure, these volumes would be sufficient in the midlatitudes, but we are dealing with the Arctic here! I see in the appendix that the observations are often at or lower than the background filters. This makes interpretation of the results extremely difficult. I am also not clear on why differential spectra were used for the blank corrections, or why there are still data points for the subtracted when the original datapoint was below the background. It is evident the authors were branching into Arctic measurements (which is great!) and were perhaps not aware of the difficult sampling conditions. Obviously, nothing can be done about this at this point, but the authors should spend the necessary time to describe these caveats in detail in the main text and CAREFULLY interpret the results given these caveats. The article is short as it stands, so certainly has the space to spend on appropriate descriptions of sampling issues that may lead to the results observed.

One of the novel aspects of this work is that the INP measurements were conducted ABOVE the ground. Arctic INP measurements are limited in general, but especially in the vertical. The authors should include more details on how these measurements were conducted exactly (e.g., which heights, etc.; see specific comments below) and include discussion and interpretation on these compared to previous ground based measurements, especially given the Arctic can be highly stratified year-round and certainly in the spring. Even if the filters suffer from high backgrounds and sampling statistical issues, this still affords key information on vertical distributions of INPs. The authors should also compare their results to previous airborne INP measurements such as those from MPACE and especially ISDAC (spring campaign in the same region).

Why is there no discussion on flying in clear air versus cloud? And if out-of-cloud, what percentage of time was spent below versus in? I would expect the results to vary quite a bit, depending on these

conditions. The authors need to include details and discussion on cloud conditions, in addition to those conditions over which surface types (ice, snow, open water, land), and the interpretation of those conditions with respect to the observational results.

**Specific comments:**

Abstract: One of the benefits of this work is INP measurements above ground, however, the abstract does not discern if there was any sort of vertical gradient in INPs. It would be important to note the altitude ranges somewhere in the abstract, and if the INPs were vertically resolved over the whole flight(s).

Lines 54-55: This is only true for the Arctic haze season in the winter/spring.

Lines 59-60: Some of these were samples collected then processed in the lab, so they _could_ contribute to the INP population should they become airborne. It is not clear if they actually do in the real environment.

Lines 60-61: What about dust sources that contain biogenic material, like permafrost? Could just reference Creamean et al. (2020) here.

Paragraph starting on line 67: Should reference the new study by Creamean et al. (2022) for Arctic interseasonal annual cycles. Specifically for the western Arctic, what about the other INP measurements in this region, including airborne studies (e.g., ISDAC, MPACE)? This is especially important to bolster since these measurements are above the ground. These references should be included here for the literature background.

Lines 72-73: Seems like this is an incomplete sentence. I assume the authors mean that Rinaldi et al., in contrast, did not observe the seasonal cycle? They actually do report a small increase in INPs from spring to summer. Important to note here too that Svalbard can be partially ice-free in Apr versus western Arctic locations, and the presence of pack ice can modulate the local sources of INPs.

Line 90: Fig 1 shows airmass backward trajectories. Where are the flight paths? Certainly important to include those somewhere.

Line 109-110: Need to describe what "handling experiment" is a bit more. I assume this means collecting a blank by handling it in the same manner as the sampled filters, but this needs to be explicitly stated.

Lines 123-139: This information belongs in the methods section, not the results and discussion.

Lines 145-146: Was this expected, based on the conditions and previous literature? Why or why not?

Lines 150 and on: The literature comparison is great, but the point to make here is that the authors measured ABOVE the ground. Given the Arctic can be highly stratified, it is challenging to compare what was measured at the ground to that aloft. The advantage of this work is that it is above ground, even if not vertically resolved. This concept needs to be highlighted, here and throughout.

Line 173: No mention of OPCs in the methods. This information needs to be included in that section (i.e., which OPCs, the inlet they were sampling on, etc.). Were the optical data corrected or quality controlled at all? If comparing the SEM and OPC data, there should have been a conversion of the different measured diameters, but I do not see this anywhere.

Section 4: The authors reference the unique properties of their samples to "other regions" they have conducted the sample analyses on, but which "other regions"? The Arctic is a very different place when it comes to aerosol sources and concentrations. It seems as if the authors have not conducted much work in the Arctic, which by no means is a problem, but doing the proper legwork is necessary here. What I mean

here is, specifically comparing to other microscopy studies in the Arctic. Kerri Pratt's group has done a number of studies using CCSEM-EDX in Northern Alaska. I strongly suggest looking up her group's papers (https://prattlab.chem.lsa.umich.edu/pubs.php) and comparing/contrasting the reported observations to those. Essentially, more interpretation of the results is needed here with respect to previous work done in the same region (albeit, at the ground).

Lines 180-181: I see the distributions for the SEM and OPC in Fig 4, but what are the percentages of particles detected from SEM via the total population in the overlapping size bins?

Lines 190-192: This could LARGELY affect interpretation of the results! But, what do the authors mean by "artefacts"? Do they mean cloud residuals, compared to interstitial aerosol? Then, were cloud particles collected via filters at times and water evaporated / ice sublimated during inlet residence time or once on the filters? See my general comment above about interstitial aerosol versus cloud residuals. Additionally, any information on the cloud phase? Temperature?

Lines 211-213: Need to cite this. But additionally, how likely is this given the airmass analysis in this region?

Lines 219 and on: Sure, this is true for lower latitudes within the dust belt, but is this really relevant for here? Why not look at trajectories farther back in time if this is a possibility? With the evidence shown, this interpretation does not fit. There are no SEM images shown, so how sure are the authors that what they are calling mineral dust is actually dust versus industrial particles from Prudhoe Bay? Did the authors evaluate the chemical spectra in the context of the morphology as well?

Lines 247-248: This cannot be confirmed without INP treatments. The authors should reword this to demonstrate this is a speculation, albeit a legitimate one.

Lines 254-257: Can the authors elaborate on this? What specifically would the biological material be from during this time of year? Should reference papers like Creamean et al. (2022) and Santl-Temkiv et al. (2019) here that do evaluate INPs/biological particles in the spring. Porter was in the late summer, so it is not exactly relevant for the Mar measurements here. Late summer sources can vary quite a bit from spring, given the contrasting transport conditions, surface open water, and marine versus sea ice biological productivity. Need to compare with previous ISDAC results (also a spring flight campaign).

Fig 1: The flight IDs are difficult to discern without the table being in the main text.

Fig 2: Starts at -14C...is this a sample limit or instrumental bias? If the former, is this because of the blanks for sample volume?

Fig 3: "All Arctic" is a misnomer. These are from Porter et al. from one study in Aug/Sep. Why are the authors showing data from other studies in the same timeframe, but from the late summer for Porter? The Porter data are not actually relevant for this figure. If the authors want to show the specific studies indicated in comparison to, truly all Arctic data, then other studies should be included from other times of the year as well.

**References:**

Creamean, J. M. et al. Thawing permafrost: an overlooked source of seeds for Arctic cloud formation. *Environ. Res. Lett.* 15, 084022 (2020).

Creamean, J.M., Barry, K., Hill, T.C.J. *et al.* Annual cycle observations of aerosols capable of ice formation in central Arctic clouds. *Nat Commun* 13, 3537 (2022).

Tina Šantl-Temkiv, Robert Lange, David Beddows, Urška Rauter, Stephanie Pilgaard, Manuel Dall'Osto, Nina Gunde-Cimerman, Andreas Massling, and Heike Wex. Environmental Science & Technology 2019 *53* (18), 10580-10590.

See link above for papers from the Pratt group for previous microscopy studies in the same region.

---

## Author Comment (AC1)

**Response to the comments**

We would like to thank all the reviews for their contributions, as we feel the quality of the manuscript has substantially improved due to their comments. We have addressed all the comments, and we have added some extra changes, that include:

- Pressure altitude has been changed to GPS altitude, which is more precise.
- Colours in the SEM graphs changed to be more intuitive and distinguishable. Bins of SEM analysis simplified, numbers of surface area have been updated (the changes are very minor).
- The back trajectories have been run as ensembles and moved to the Supplementary Information.

RC1

General comments
1. A bit more discussion of aerosol-cloud interaction is needed. I.e. what do the results mean for mixed phased clouds and the overall INP "budget" in this region?
*We held off extending this paper to a study of aerosol-cloud interactions and purposefully kept it to a study of the INP and aerosol populations in the springtime Arctic. In the case of this particular study, we piggy backed on a campaign and made opportunistic measurements of INP and aerosol, so there was minimal opportunity to link our results to cloud properties. In more recent campaigns conducted in 2022 we have won funding to define aerosol-cloud flights where the INP we measure are of direct relevance to cloud systems we probed in the same flights.*

2. The authors suspect low-latitude sources for the mineral dust, is it possible to be more specific in that regard? It is known that mineral dust sources have compositional fingerprints (Scheuvens & Kandler 2014, and references therein, https://doi.org/10.1007/978-94-017-8978-3_2). Could the authors use the SEM-EDS results to constrain possible source regions? Could the authors also use satellite products to at least qualitatively verify the transport of dust from low to high latitudes over the relevant time period?
*SEM: Unfortunately, so few particles were observed due to the low concentrations that the statistics are poor. However, the analysis in Sanchez-Marroquin et al. (2020) shows that the Alaskan dust looks more like Barbados dust than Icelandic dust. This does not mean the dust is from Africa, but it probably isn't from a location with similar mineralogy to Iceland. A more complete study would be interesting, but beyond the scope of what is feasible.*
*Using satellite products: The dust concentrations we are talking about here are very low and do not show up in typical satellite products. We are not talking about a well-defined plume, but instead the pervasive accumulation mode dust that has a lifetime of many days to months in the atmosphere.*

3. Several details on the flights and the sampling strategy are missing in my view, but are needed for a reader to assess the conditions in which the samples were collected. Hence I suggest the addition of the following information to the manuscript (at least in the appendix) • Individual Flight tracks and/or height profiles with the sections when a samples was collected highlighted.
• Indicating if a samples was collected within the PBL/MBL or in the free troposphere; Indicating if sample was collected below/above a cloud layer. This could be done as additional columns in Table A1
• If available, vertical profiles of aerosol number concentration (e.g. derived by integrating the OPC size distribution) and basic meteorological parameters (temperature, humidity)

*The referee makes some good suggestions here. We have added the mean temperature, dew point and number concentration (for particles 0.1 to 3µm) into Table 1, which is now in the main text. PBL/MBL has also been added to the table. Flight tracks have been added to Fig. 1.*

4. For details on the inlet, the authors refer to one of their previous publications, which is fine, but nevertheless the main key should be repeated in this manuscript. That includes cut-off diameter, is the inlet heated or is the aerosol stream dried in some way to avoid the collection of cloud droplets, is the sampling isokinetic, and where on the aircraft is the inlet located?
*We have added some pertinent details of the inlet system in the methodology.*

5. Analogous to my previous comment, the main key points of the SEM-EDS should be repeated in this manuscript and not just referenced. This includes whether there was any form of sample preparation, such as gold sputter coating, what accelerating voltage was used, how long was the collection time for an EDS spectrum, whether the analysis was manual or automatic, (if the
latter, whether particle detection was based on contrast in the BSE image or something else, and whether possible artifacts were removed before further analysis).
*We have added some pertinent details of the SEM technique in a concise manner to help the reader understand what we have done.*

L64 Since Tobo et al. 2019 has already been cited above, and because it is consistent with one of your conclusions, it should be explicitly mentioned that dust can also be a carrier for biological INPs
*We have added an explicit statement in the introduction: 'Biogenic material attached to dust particles could be an important part of these terrestrial INPs (O'Sullivan et al., 2014;O'Sullivan et al., 2015;Tobo et al., 2019).'*

L72-73 This statement about Rinaldi et al. in relation to Wex et al. is correct, but also a bit misleading. Wex et al. attribute the seasonal increase to biological INP and see their effect primarily at high temperatures, with a less pronounced seasonality at lower temperatures. Because of the setup used by Rinaldi et al. they can measure at lower temperatures/higher concentrations. So even if there were a seasonality, Rinaldi et al. would not see it as pronounced in their data, as Wex et al. do. The wording here should take this into account.
*The measurement range of Rinaldi and Wex do overlap, with Rinaldi's between $10^{-3}$ and $1\ L^{-1}$ and Wex's between $10^{-4}$ and $3x10^{-2}\ L^{-1}$. However, the range of dates over which measurements were made by Rinaldi is smaller than Wex and we should mention this. The pertinent sentence has been adjusted to: 'However, a recent study did not find strong seasonality of Arctic INPs at Ny-Ålesund, although their measurements were limited to being between April and August 2018'*

L116-117 What is meant by "subset"? Were not all PC filters used for SEM-EDS? If so, what happened to the rest and how was this "subset" selected?
*Not all the PC filters were analysed using SEM-EDS for two main reasons. One reason is that the aerosol concentrations were very low, hence samples with low sampling volumes did not have enough particles for the analysis. The second reason is that the technique is very time-consuming and the main author lost access to the SEM-EDS facilities when the COVID-19 pandemic started. Therefore, only a selection of samples could be analysed before the project finished. The criteria to choose samples was getting a sample with a relatively large sampling volume from each flight.*

L153-155 I would suggest also including a more recent publication on aircraft INP measurements in the Arctic such as Hartmann et al. (2020), which you cited earlier, in the comparison. They differ in location but are very similar in time (March/April 2018). Borys (1989) may be more comparable in location, but his measurements took place more than 30 years ago, before the extraordinary warming of the Arctic was observed (Arctic Amplification), and some might argue that the differences are related to this.

*Our objective here was to first specifically to compare our results to measurements in close locations and timings, and then comparing our measurements to the range of all INP measurements carried out in the Arctic (which includes the Hartmann dataset). We have split figure 3 so now it includes a detailed comparison to the measurements in a close by location and a full comparison to all the existing Arctic measurements.*

L155 When making the comparison here, it should also be mentioned which method was used in the respective studies to determine [INP] (freezing assay, CFDC, expansion chamber).
*Added to Sec. 3.*

L172 There are many ways to obtain particle diameters in SEM images, it should be mentioned what diameter was used here
*Added to Sect. 4: "The equivalent circular diameter size distributions obtained with the SEM-EDS technique were compared with…"*

L173 More details about the optical probes are needed, such as the name of the instrument, the manufacturer, and where it is mounted. Before reading in the caption of Fig. 4 that it was a PCASP-CDP, I wondered if the OPC might have been fed through the same inlet as the filter sampler. PCASP-CDP, on the other hand, indicates that it is a wing-mounted device. In addition, it should be mentioned whether the OPC size distributions shown in Fig. 4 represent the average over the entire period in which the SEM sample was taken or whether it is one point measurement during that sampling period.
*These OPCs were all under-wing probes, so not behind an inlet. We have added some details at the beginning of section 4.*

L175 (+ Fig 4) Is there a reason why only four samples were analyzed with SEM-EDX? An explanation should be given in the text.
*We addressed this above.*

L190-192 PCASP-CDP suggests the combination of aerosol and cloud probe. Doesn't this make it possible to confirm rather than speculate whether the artifacts are actually cloud droplets?
*We could not confirm that fully for two reasons. The first is that the shape of the artefacts is rather a-physical, as the concentration decreases more than four orders of magnitude in about 5 µm. This does not look like a log normally distributed size distribution of cloud droplets. The second reason is that our sampling strategy deliberately avoided sampling in cloud and rain conditions, so we do not expect a very large amount of droplets detected across the run.*

L201 Because of the scale of the y-axis in Fig. 1b, it is difficult to tell when the air masses are below 500m as described in the text. A horizontal line indicating the 500m mark might help, and perhaps a logarithmic axis.
*Done*

L204-207 (+ Fig 1a) Looking at sea ice concentration (e.g. https://seaice.uni-bremen.de/databrowser/#day=11&month=2&year=2018&img=%7B"image"%3A"image-1"%2C"product"%3A"AMSR"%2C"type"%3A"nic"%2C"region"%3A"Arctic3125"%7D ) instead of sea ice extent or satellite imagery in general ( e.g., https://go.nasa.gov/3TevAJn ), it is clear that the ocean around the north coast of Alaska is not completely covered by ice, but is highly fragmented with numerous open leads. This makes the marine particle source mentioned in L207 more relevant in my eyes and should be discussed accordingly in the text.
*We did not intend to rule sea spray INP production out and the referee is correct in stating that there were open leads. Other work from our group indicates open leads at the North Pole were a weak source of INP, but not necessarily negligible, and also this was the North Pole not the Alaskan Arctic*

*which might be different. We have emphasised this possibility in Sect. 4: "Hence, it is possible that the sea spray particles had been emitted from open leads in the sea ice, as this is thought to be a common source of sea spray aerosol in the region {May, 2016 #873;Kirpes, 2019 #758;Chen, 2022 #869}."*

L210-211 Defining mineral as described here is reasonable in many cases. However, considering that there are algae with Si or Ca-based skeletons and the question of the importance of marine or terrestrial INP sources is still debated, I wonder if algal skeletal fragments were misclassified as mineral dust. Can the authors comment on this? Was visual screening done to determine the amount of non-dusty Si and Ca containing particles?

*Although the majority of the dust is in the Al-Si rich category (contains Al) or Si-rich categories (contains other elements apart from Si other than Ca), there is a chance that algae skeleton fragments could end up in the Si rich or Ca rich categories, as in any other dust measurement done with this technique (or mass spectroscopy). In this study, a qualitative visual inspection of some of these particles did not reveal any diatom structures, but we cannot categorically rule them out.*

L219-222 Indeed, studies such as Huang et al. (2015) show that dust from lower latitudes can reach the Arctic within 5 days. The back trajectory analysis in this study also states that the air masses containing the dust particles circulated for at least 5 days. Meaning that the found mineral dust particles with a size of 1-5 µm had to remain suspended in the air for a minimum of 10 days. Can the authors support the claim this typically happens in the atmosphere?

*The residence time of dust aerosols in the atmosphere can range from a few days to weeks (Di Biagio., 2021 10.1016/B978-0-12-818234- 5.00033-X, Uno et al., 2009; Huneeus et al., 2011; Ménégoz et al., 2012) and in the Arctic this can be months (Carslaw 2022). Additionally, not all the air masses had been circulating around the Arctic; a couple of them came from the south, and some mixing can occur.  We have added a short statement to this effect in the SEM discussion section 4:*

*"Almost all the mineral dust particles found in this study had sizes below 5 µm and it is known that the lifetime of accumulation mode aerosol typically has a liftime of many days or weeks so can be transported to Alaska from distant sources {Huneeus, 2011 #637; Ménégoz, 2012 #835}.  Once in the Arctic, accumulation mode aerosol has a lifetime extending to months during winter and spring, when removal processes are weak (Carslaw 2022)"*

L234 Fig. 5 (instead of Fig. 3)?
*Thanks for spotting the typo.*

L235-236 Is it not possible to use the SEM-EDS analysis to derive a K-feldspar amount that is more representative of the samples of this study? You already have the "Al-Si rich" category, which should contain mainly the feldspars and their weathering products (illite, kaolinite, etc.). If this category is filtered for appropriate amounts of K, an estimate for the K-feldspar content could be derived. I think studies by K. Kandler and co-authors have covered the categorization of mineral dust with SEM-EDS in detail.

*Yes, we tried the approach by Kandler et al, however, the number of dust particles analysed in this study is relatively low due to the low aerosol concentrations of the Arctic (always under two thousand per filter), so the statistics of K-feldspar particles were even lower, hence we do not report them.*

L238-242 It should be noted that the parameterization presented by McClusky et al. (2018) is meant to represent pristine SSA and they intentionally excluded events with elevated INP concentrations. They also state t during a period of elevated marine organic aerosol from offshore biological activity, [INP] at -15°C was $7.7 \times 10^{-3}$ $L_{-1}$. A value that is above the mineral dust and below the fertile soil in Fig. 5. This, of course, does not refute the author's conclusion that ice activity is related to mineral dust, but the possibility of biological INP of marine origin should be discussed. McClusky et al. (2018) also present a different parameterization based on TOC. Perhaps the authors can derive an estimate for

TOC based on the "carbonaceous" category of the SEM-EDS data and add these predicted INP concentrations to Fig. 5 as well.

*This is a good point regarding the McCluskey data. We have adjusted the discussion to take this into account and now state: 'It is possible that the sea spray in this location was more active than defined by McCluskey et al. (2018), however, the INP concentrations calculated based on the presence of dusts better explains the observed INP concentrations' and further down we now state 'hence this suggests that the samples from Alaska contained some biological ice nucleating material (either from marine or terrestrial sources).*

*Unfortunately, the carbonaceous category in the SEM analysis only indicates that those particles did not contain any other major element apart from the elements in the filter (carbon and oxygen), but we cannot quantify anything else with them. Additionally, some of the TOC would come in the sea spray particles, which are classified as Na rich. The fact that the filters are made of carbon and oxygen means we detect those elements everywhere in the analysis and we cannot quantify much of these elements.*

L252-254 Following the author's train of thought, the sampled aerosol was transported for about 10 days, and as noted here, a decrease in ice activity would have been expected, yet Fig. 5 shows a very good match with fertile soils collected directly from the ground and brought to the laboratory quickly. This would suggest that the aerosol was not transported far, but may have more local sources. Can the authors comment on that? In this context, could the use of monthly average snowpack (Fig. 1a) or the sensitivity of the satellite product used for the ERA5 reanalysis be obscuring exposed soils?

*In our opinion it is not obvious that there should be a decrease in activity on transport. While it has been shown in the lab that mixing with acids, for example, can deactivate some INP types it is not clear that this process occurs in the atmosphere to a significant extent.*

L255-256 In the INP community, heat and $H_2O_2$ treatments are commonly used to gain additional insight into the presence of proteinaceous and organic INPs. Could the authors perform such treatments as the results could further support the claim presented here?

*Unfortunately, the droplet-on-filter INP measurement technique only allows us to perform the analysis once, hence, heat tests or $H_2O_2$ treatments cannot be done in our samples. Also, it is not clear how the filter would respond to these treatments (although we have considered it and recognise the value). This is a disadvantage of the droplet-on-filter technique, but the greater sensitivity over the wash-off methods means we can make INP measurements in the 20 minutes or so that we have per filter leg. Hence, this compromise is well worth it.*

Table A1, second column 2018 instead of 2017
*Corrected.*

Table A1 Are the presented altitude values averages over the sampling time? If yes, that should be mentioned in the text and caption.
Added to the caption of the table : "The given altitude values correspond to the average of each run."

Fig. 1 a) A reader is not able to recognize where a trajectory starts/ends. If the daily marker would shrink with time, the reader might be able to identify the direction of the trajectory and hence start and end. The authors also chose to show only one return trajectory per sample. For short sample times, a single back trajectory could be representative, but for longer sample times, the aircraft could travel long distances and enter air masses with very different histories. Can the authors comment on the representativeness of back trajectories they show. The authors might also consider adding a figure to the appendix showing high frequency back trajectories for each sample.

*The fight tracks have been added to figure 1, while the back trajectory analysis has been sent to the supplementary material. The analysis has been separated per days to ensure it is possible to see each trajectory better and the single trajectories have been substituted by ensembles.*

*We have not performed a more detailed back trajectory analysis.  We considered investing more time in a Flexpart study, but since the indication is that the sources are not local, we suspect there is little to be gained from a more detailed back trajectory approach. Also, our sampling periods tended to be focused on specific air masses, rather than sampling across air masses or fronts, hence we consider them to be qualitatively representative.  In addition, detailed back trajectories would be required if we saw some filters with much greater INP concentrations than others, but this was not the case.  Hence, we just didn't observe sufficient natural variability for source apportionment modelling.*

---

## Author Comment (AC2)

**Response to the comments**

We would like to thank all the reviews for their contributions, as we feel the quality of the manuscript has substantially improved due to their comments. We have addressed all the comments, and we have added some extra changes, that include:

- Pressure altitude has been changed to GPS altitude, which is more precise.
- Colours in the SEM graphs changed to be more intuitive and distinguishable. Bins of SEM analysis simplified, numbers of surface area have been updated (the changes are very minor).
- The back trajectories have been run as ensembles and moved to the Supplementary Information.

RC2

General comments:
1. Better to give more detailed information about the flight track and sampling strategy. For example, Appendix A shows the sampling time varies from ~10 to 30 mins. During the sampling, does the flight sample cloud or not? The flight height varies from 100 to 700 m, how much percent of the time is within the boundary layer of each sample?
Extra information about the samples has been added in the table (above or below boundary layer) and some extra description of the sampling strategy (all sampling was done outside cloud) has been added to the first paragraph of section 2. Additionally, flight tracks have been added to fig. 1.

2. The March in the Arctic is very often influenced by Arctic haze (not only long-range transport of dust but also anthropogenic pollution or biomass burning). Do you see any indication of anthropogenic pollution? For example, sample C089_3 shows a high carbonaceous fraction, what is the source of carbonaceous? Why only reported the chemical composition of four samples?
In our SEM analysis, carbonaceous particles are the ones that do not contain any element beyond the ones present in the polycarbonate filters (carbon and oxygen). Because of that and the fact that most carbonaceous combustion particles are at the limit of detection of the technique (and many of them are missed by it, as explained in Sanchez-Marroquin et al, 2019), we cannot say much about combustion particles. Additionally, our INP analysis technique is also not very sensitive to carbonaceous particles as these ones are not very effective at nucleating ice (Vergara-Temprado et al., 2018, Adams et al., 2020). Hence, we are not in the position of saying much about this type of particles.
Only four samples were analysed using SEM-EDS for two main reasons. One reason is that the aerosol concentrations were very low; hence, samples with low sampling volumes did not have enough particles for the analysis. The second reason is that the main author lost access to the SEM-EDS facilities when the COVID-19 pandemic started. Therefore, only a few samples could be analysed before the project finished. The nature of the analysis is that each sample takes about 1 day of SEM time and post-analysis and caries a substantial cost, hence it is not something that can just be squeezed in. The criteria to choose samples was getting a sample with a relatively large sampling volume from each flight.

Minor comments:
1. Lines 67-79: It is better to include the recently published paper from Creamean et al. (2022), which discussed the seasonal variation of INP in the central Arctic.

We have redrafted this whole paragraph to better take into account the literature data, including the contrast between the Creamean 2022 and the Porter 2022 conclusions.

2. Line 98: Add "open leads" after snow. Open leads are most likely omnipresent in the central Arctic. And later when you discuss the aerosol sources in Lines 201 -209, the sea salt aerosols are also likely from open leads.

We added "However, local sources of marine aerosol particles may still occur due to open leads {May, 2016 #873;Kirpes, 2019 #758;Chen, 2022 #869}." At the end of Sec. 2.

3. Line 137: Change "approach to (Vali, 2019)" to "approach to Vali, (2019)".

Thanks for spotting.

4. Lines 190-192: Do the aircraft measurements also have cloud-related measurements, like liquid water content or cloud droplet number concentration?

The focus of these flights was not on aerosol-cloud interactions, hence cloud measurements are very limited and we therefore do not report them here. As mentioned in response to referee 1, we took advantage of these flights in an opportunistic manner. In 2022 we used the same technique in flights designed to measure aerosol and INP relevant for specific clouds.

---

## Author Comment (AC3)

**Response to the comments**

We would like to thank all the reviews for their contributions, as we feel the quality of the manuscript has substantially improved due to their comments. We have addressed all the comments, and we have added some extra changes, that include:

- Pressure altitude has been changed to GPS altitude, which is more precise.
- Colours in the SEM graphs changed to be more intuitive and distinguishable. Bins of SEM analysis simplified, numbers of surface area have been updated (the changes are very minor).
- The back trajectories have been run as ensembles and moved to the Supplementary Information.

RC3

**General comments:**
The introduction is too short and does not include enough relevant background. For instance, there is only a very brief introduction to Arctic mixed-phase clouds here. Some more effort should go into describing AMPCs, INP observations, and how limited they are relative to other latitudes. For the INP aspect, since this is an Arctic manuscript, the introduction material on INPs should focus more on what has been observed in that region. There are some missing key references and studies of Arctic INPs. See my specific comments below on these and what should be included. In general, *some* of the relevant literature is included, but should be more complete and involve discerning where and when the observations were made, since those can vary quite a bit depending on location, sea ice extent, etc. This is hinted at on lines 150-151, but should be elaborated on in the introduction to develop the broader picture for the reported observations.

We have expanded the discussion of previous INP research in the Arctic and also included a specific mention of Arctic mixed phase clouds in the introduction. We do not wish to go into great depth on the arctic mixed phase clouds because we did not actually study them, but rather focused on the aerosol. We disagree that measurements are particularly 'limited' in the Arctic relative to elsewhere. There is a dearth of INP measurements around the globe. Regarding the comment on being 'more complete and involve discerning where and when the observations were mode: We did exactly this and state that we compare our data to ' literature data collected in a similar location and time of the year in Fig. 3'. We then go on to set the data in the context of a compilation of Arctic data (Fig. 3b); this is now much clearer in the revised figure 3.

Even though some of the methodologies are described elsewhere, pertinent information should at least be distilled here so the readers do not have to go on a wild goose chase to obtain such information. Much of it is buried in the appendix, but why not include it in the main text? Here are some questions I had after reading through the methods, information that could be briefly added for clarity:

Overall, we have expanded the explanation of all the used methods through the text.

● When I first read it, it was not clear if only 1 filter was collected per flight, for how long, which altitudes, etc. Table A1 has this information, which I think would be important to include in the main paper.

*The methodology has been expanded in the main text, the table A1 has been moved to the main paper.*

● There is not enough information on filter collection, preservation, and analysis in the main text. Again, this information is in Table A1 and its caption. Should be moved to the main text. For the altitude, was the aircraft spriling at that one altitude per filter? If so, what was the spatial coverage of

these spirals? Or were these profiles and the one altitude value provided is an average? Need more details here.

*The flight tracks have been added to Fig. 1. The table is now in the main paper.*

● I originally assumed the processing was not done on site, however processing was indeed done on site as stated in the caption for Table A1. This may seem like mundane information, but how the filters are preserved and handled can have a large impact on their results, so is important enough to include in the main text.

*We have added a few lines on the advantages of doing this analysis on-site.*

● What filter pore size was used for the filters? Different filters have different collection transmission efficiency curves.

*Information about the filters and a reference to our discussion on filter collection efficiency has been added to the second paragraph of Sect 2.*

● Why was Teflon used for INPs? Teflon tends to have high backgrounds compared to polycarbonate. And why polycarbonate for microscopy?

*We have added a few lines on why we used it: A disadvantage of the droplet-on-filter technique is that each sample can only be analysed once, which makes it incompatible with standard heat tests, for example. However, the great advantage of the droplet-on-filter technique over techniques where particles are washed off a filter into a volume of water is that it is around 20 times more sensitive than a typical wash-off assay employing 1 μl droplets (depending on the details of the freezing assays). This enhanced sensitivity is very important given that aerosol sampling durations are typically only a few tens of minutes long.*

*Polycarbonate filters were used for SEM because these give a flat surface on which we can identify particles.*

. How were carbonaceous particles classified if the filter substrate material is carbon-rich?

*Polycarbonate were used for SEM-EDS analysis. Particles were detected based on their morphology, and they were classified carbonaceous if they do not contain further elements beyond C and O, the elements present on the background filter. We have expanded the section on the SEM, but refer to the original paper regarding the details of the classification scheme.*

● How exactly were the sample suspensions prepared? Shaken in water? For how long? What type of water was used? And was a blank water spectrum tested as well to see how much of the background is from the water itself versus the filter handling?

*INP concentrations were analysed using a droplet-on-filter technique, explained in the second paragraph of the Sect. 2 (i.e. an advantage of this technique is that we did not prepare a suspension). Blanks and handling blanks were performed, as explain on this paragraph and shown in Fig. 2a.*

● How many drops and how many separate tests for the drop freezing experiments?

*We have added this information to the section 2: "On average, we pipetted 54 (with and standard deviation of 5) droplets per filter."*

● What type of cold stage is this? Since it is not a commercial instrument, there needs to be some description of the method, even if just a few sentences.

*Additional details have been added to the third paragraph of Sec. 2. We note that we have published papers with this cold stage for over a decade and citations to details are given.*

● What is the temperature range of the cold stage?

*The cold stage can work at temperatures within the range of 30 to -100°C, but the experiments were performed at a range of ~-10 to -35 °C.*

● Was freezing detection manual (by human eye) or automatic?
*The videos of the freezing were manually analysed. This has been added to the text.*

● Why was a 68% confidence interval chosen? That seems like an oddly low one…
*The important thing here is to report what confidence interval we report the data to. We chose to report it to 1 standard deviation (i.e. 68%) to be consistent with our previous work.*

● I realize the inlet is described elsewhere, but clear referencing is needed, given aircraft inlets can be riddled with artefact issues.
*Added to second paragraph of Sec. 2.*

● For HYSPLIT, why were the heights and 5-days chosen? A topic of discussion in the community is how long back is long enough. I don't have a good answer to that, but the authors should at least justify why they chose the parameters that were used. Additionally, typically a good approach is to run trajectory ensembles (i.e., every so many hours during filter collection and at multiple heights around the sampling point). One trajectory per sample is not enough, given the uncertainties with HYSPLIT, especially near the poles.
*We have moved the back trajectory analysis to the supplementary material, and run trajectory ensembles for each sample. In terms of why 5 days we are aware that confidence decreases the further back in time we go.*

● There is hardly any information on the microscopy work. Even though it is in a previous paper, a few more sentences on the methodology should be included here. Where was the SEM-EDX analysis done? What about the details of how the samples were processed on the same filters? Were the samples frozen or stored at room temp and why? There should also be a brief synopsis of how the classifications were defined.
*A more detailed description of the SEM-EDS technique has been added in response to referee 1. Some more information about the S rich and Carbonaceous categories has been added to the third paragraph of Sect. 4, alongside the existing information on the Na rich and dust categories.*

These are just a handful of questions I had, but effectively, the authors should take care to include the relevant methodology information in a succinct manner in the main text instead of in the appendix. That would alleviate most of my concerns here. The details on the  calculations and blank corrections could remain in the appendix, if the authors desire, but the rest would be more helpful in the main text.
*We have substantially improved the amount of information about the methodology throughout the text. We hope this makes the reading more clear.*

I am concerned about the background INP spectra. These are very high, and I suspect part of it is due to the use of Teflon and the fact that only a few hundred liters of air were collected per filter.
*We are aware the Teflon sometimes have a higher background, however, as stated above the droplet-on-filter technique has a greater sensitivity.  We cannot use the polycarbonate filters for the droplet-on-filter technique because they are not sufficiently hydrophobic. In addition, we find that the background is defined by the environment that we work in as much as the filter type.*

Sure, these volumes would be sufficient in the midlatitudes, but we are dealing with the Arctic here!
*We disagree with implications of this statement.  The volume of air sampled is limited by the length of time we can sample for. This is dictated by the demands on time within the aircraft sorties.  The total*

*duration is four hrs including transit, hence a few filter runs of 20 minutes each is what we have to work with. The results we produced are still meaningful, despite many being upper limits. Aircraft sampling is very challenging for many reasons and a great deal of hard and valuable work should not be dismissed with statements like the referee's.*

I see in the appendix that the observations are often at or lower than the background filters. This makes interpretation of the results extremely difficult.
*That is correct, we are only able to report upper limits for many of our measurements on this campaign because the INP concentrations in this location at this time were remarkably low. Nevertheless, upper limits are still very useful quantities and the results are treated and discussed accordingly.*

I am also not clear on why differential spectra were used for the blank corrections,
*The background subtraction must be done with differential spectrum (Vali, 2019).*

or why there are still data points for the subtracted when the original datapoint was below the background.
*We presume the referee is referring to the cumulative spectra. One has to remember that each point in the cumulative spectra is the integration of the differential quantity between 0°C and T. Hence, even if the differential quantity is zero at temperature T, the cumulative quantity will not be zero if the differential quantity is larger than zero in any of the points at greater T. For example, if we are confident that the INP concentration at -20 °C is 0.1 L$^{-1}$ (because the cumulative k is statistically above the limit of detection), then the INP concentration at -21 °C cannot be smaller than this value, regardless of what the k value at -21 °C is. This is because a cumulative spectrum cannot decrease with decreasing temperatures. This is already explained in the last paragraph of the Appendix 1 and is detailed extensively by Vali (2019).*

It is evident the authors were branching into Arctic measurements (which is great!) and were perhaps not aware of the difficult sampling conditions.
*We would like to point out that we have conducted a number of campaigns in the Arctic. We disagree with the idea that measurements in the Arctic are necessarily more challenging than elsewhere. This referee comment seems to be based on the notion that INP concentrations in the Arctic are always low, but this is far from the fact. For example, in our recent paper on measurements from the North Pole (Porter et al., 2022) we observed INP concentrations ranging from those representative of the remote Southern Ocean to as high as mid-latitude terrestrial locations, hence any sampling method and analysis needs to capture as much of this as possible, which is what we have aimed at.*

Obviously, nothing can be done about this at this point, but the authors should spend the necessary time to describe these caveats in detail in the main text and CAREFULLY interpret the results given these caveats. The article is short as it stands, so certainly has the space to spend on appropriate descriptions of sampling issues that may lead to the results observed.
*We have expanded on a number of points raised by this and other referees.*

One of the novel aspects of this work is that the INP measurements were conducted ABOVE the ground. Arctic INP measurements are limited in general, but especially in the vertical. The authors should include more details on how these measurements were conducted exactly (e.g., which heights, etc.; see specific comments below) and include discussion and interpretation on these compared to previous ground based measurements, especially given the Arctic can be highly stratified year-round and certainly in the spring.
We have expanded the introduction to include the following 'Given there are strong aerosol sinks in the boundary layer, whereas the air above the boundary layer can be stratified with corresponding long aerosol lifetimes (Carslaw, 2022), vertical measurements are required. Hartmann et al. (2020)

report INP spectra for late March and early April north of 80° over the Fram Straight and Arctic Ocean and report that the highest INP concentrations ($2 \times 10^{-2}$ L$^{-1}$ at -15°C) correspond to the boundary layer, indicating a local marine source even though the region was mostly ice covered.'

Even if the filters suffer from high backgrounds and sampling statistical issues, this still affords key information on vertical distributions of INPs. The authors should also compare their results to previous airborne INP measurements such as those from MPACE and especially ISDAC (spring campaign in the same region).

We have included information on altitude and if the measurements was in the free troposphere or boundary layer in Table 1. The number of measurements clearly in the boundary is low, hence it is not possible to make statements about trends. The nature of the filter based technique means that it is not possible to build plots of vertical distributions of INP. This is where online instruments, such as CFDCs have an advantage.

In terms of the discussion of the relationship with other field data we have purposely focused on datasets in a similar region at the same time of year. A secondary focus is then showing how our results fit with the broader range of INP measurements that together indicate a massively variable INP concentration.

We have added additional data to the comparison in Figure 3, including data from ISDAC and MPACE. Published data from ISDAC is unfortunately limited to just a few case studies (although there is more data on the ARM data repository, information on the context of this data is limited in the literature, hence we decided to limit what we present here to the published data).

Why is there no discussion on flying in clear air versus cloud? And if out-of-cloud, what percentage of time was spent below versus in? I would expect the results to vary quite a bit, depending on these conditions. The authors need to include details and discussion on cloud conditions, in addition to those conditions over which surface types (ice, snow, open water, land), and the interpretation of those conditions with respect to the observational results.

We do not sample aerosol in clouds or in regions of precipitation. The inlet is not designed to do this. This is now clearly stated in section 2.

**Specific comments:**

Abstract: One of the benefits of this work is INP measurements above ground, however, the abstract does not discern if there was any sort of vertical gradient in INPs. It would be important to note the altitude ranges somewhere in the abstract, and if the INPs were vertically resolved over the whole flight(s).

The altitude range has been included in the abstract.

Lines 54-55: This is only true for the Arctic haze season in the winter/spring.

The cited references clearly show transport of low latitude dust throughout the year. No change.

Lines 59-60: Some of these were samples collected then processed in the lab, so they *could* contribute to the INP population should they become airborne. It is not clear if they actually do in the real environment.

The cited paper, Sanchez-Marroquin (2020), is a study where dust was sampled from the aircraft where the dust from Iceland clearly had become airborne. No change made.

Lines 60-61: What about dust sources that contain biogenic material, like permafrost? Could just reference Creamean et al. (2020) here.

Yes, this is another possibility. We have added this reference to the list of citations.

Paragraph starting on line 67: Should reference the new study by Creamean et al. (2022) for Arctic interseasonal annual cycles. Specifically for the western Arctic, what about the other INP measurements in this region, including airborne studies (e.g., ISDAC, MPACE)? This is especially important to bolster since these measurements are above the ground. These references should be included here for the literature background.

This important new reference has been added in the revised text: 'Similarly, year-round measurements in the central Arctic indicate peak concentrations in the summer months of 2020 (Creamean, 2022). Creamean et al. (2022) suggested that local Arctic marine sources might contribute to the elevated INP populations in the summer.'

Lines 72-73: Seems like this is an incomplete sentence. I assume the authors mean that Rinaldi et al., in contrast, did not observe the seasonal cycle? They actually do report a small increase in INPs from spring to summer. Important to note here too that Svalbard can be partially ice-free in Apr versus western Arctic locations, and the presence of pack ice can modulate the local sources of INPs.

We have edited this sentence to read 'However, a recent study did not find strong seasonality of Arctic INPs at Ny-Ålesund, although these measurements were limited to being between April and August 2018'.

Line 90: Fig 1 shows airmass backward trajectories. Where are the flight paths? Certainly important to include those somewhere.

The flight tracks have been included.

Line 109-110: Need to describe what "handling experiment" is a bit more. I assume this means collecting a blank by handling it in the same manner as the sampled filters, but this needs to be explicitly stated.

This is defined in the methods section.

Lines 123-139: This information belongs in the methods section, not the results and discussion.

We have taken the referee's advice and moved this to the methods.

Lines 145-146: Was this expected, based on the conditions and previous literature? Why or why not?

A discussion of how it fits with the literature comes in the next paragraph.

Lines 150 and on: The literature comparison is great, but the point to make here is that the authors measured ABOVE the ground. Given the Arctic can be highly stratified, it is challenging to compare what was measured at the ground to that aloft. The advantage of this work is that it is above ground, even if not vertically resolved. This concept needs to be highlighted, here and throughout.

We have modified the text to stress that our sampling was airborne in the text and also added this brief discussion: 'Given the variability in INP concentrations and the limited amount of data we cannot say anything about the vertical distribution of INPs in this location. Given the atmosphere is highly stratified, it would be interesting to perform simultaneous measurements at the surface and from an aircraft to explore how INP at the surface are related to those higher in the boundary later and those in the free troposphere might be related.'

Line 173: No mention of OPCs in the methods. This information needs to be included in that section (i.e.,which OPCs, the inlet they were sampling on, etc.). Were the optical data corrected or quality controlled at all? If comparing the SEM and OPC data, there should have been a conversion of the different measured diameters, but I do not see this anywhere.

We have added a section on this in the methodology

Section 4: The authors reference the unique properties of their samples to "other regions" they have conducted the sample analyses on, but which "other regions"? The Arctic is a very different place when it comes to aerosol sources and concentrations. It seems as if the authors have not conducted much work in the Arctic, which by no means is a problem, but doing the proper legwork is necessary here. What I mean here is, specifically comparing to other microscopy studies in the Arctic. Kerri Pratt's group has done a number of studies using CCSEM-EDX in Northern Alaska. I strongly suggest looking up her group's papers (https://prattlab.chem.lsa.umich.edu/pubs.php) and comparing/contrasting the reported observations to those. Essentially, more interpretation of the results is needed here with respect to previous work done in the same region (albeit, at the ground).

We primarily use the size resolved composition to say something about the INP population, rather than a discussion of the composition more generally. Also, we are not aware of composition measurements at the same time of year as ours. We have added a reference to Gunsch from Pratt's group: 'This is consistent with other SEM-EDS studies of the aerosol samples collected on the Alaskan Arctic from the ground {Chen, 2022 #869;Creamean, 2018 #604;Kirpes, 2018 #871;Gunsch, 2017 #870} or during a ship campaign {Kirpes, 2020 #872}.'
We have been more specific about what we mean by 'other locations': 'around Iceland, the eastern tropical Atlantic and the south east of the United Kingdom'.

Lines 180-181: I see the distributions for the SEM and OPC in Fig 4, but what are the percentages of particles detected from SEM via the total population in the overlapping size bins?
We refrain from doing this. The temptation would be to say one measurement is the standard and one is therefore biased. Both methods have several caveats and assumptions, hence we leave it is a comparison.

Lines 190-192: This could LARGELY affect interpretation of the results! But, what do the authors mean by "artefacts"? Do they mean cloud residuals, compared to interstitial aerosol? Then, were cloud particles collected via filters at times and water evaporated / ice sublimated during inlet residence time or once on the filters? See my general comment above about interstitial aerosol versus cloud residuals. Additionally, any information on the cloud phase? Temperature?
This discussion is about the OPCs, not the filters. This does not affect the interpretation of the results 'largely'. Since submitting this paper we have been working on other datasets from the CDP and also looked back at some older datasets and it appears that there is something we don't fully understand in these size distributions. This is ongoing work and we have adjusted the text to reflect the possibilities: 'However, for samples C087_1 and C091_2, the optical counters detected a larger concentration of particles with sizes ~5 to 10 µm than the SEM analysis of the filters. Similar discrepancies have been observed previously with these instruments (Young et al. 2015) and were attributed to regions of high humidity even if the average humidity in a run should not have led to substantial hygroscopic growth. We also note that the instrument that reports size distributions in this range is designed for cloud droplets, and we are using it at the edge of its capability for larger aerosol particles, hence there may be some biases. Another possibility is that there are losses in the filters inlet system not accounted for by Sanchez-Marroquin et al. (2019).'

Lines 211-213: Need to cite this. But additionally, how likely is this given the airmass analysis in this region?
We would prefer to leave this. It is not inconceivable that combustion or volcanic ashes are present in this region.

Lines 219 and on: Sure, this is true for lower latitudes within the dust belt, but is this really relevant for here? Why not look at trajectories farther back in time if this is a possibility? With the evidence shown, this interpretation does not fit. There are no SEM images shown, so how sure are the authors

that what they are calling mineral dust is actually dust versus industrial particles from Prudhoe Bay? Did the authors evaluate the chemical spectra in the context of the morphology as well?

Many studies show mineral dust from low latitude does make it into the Arctic (including those cited in the paper). Furthermore, the lifetime of aerosol in the Arctic is very long because the sinks are weak and the atmosphere is stratified.

We didn't look at trajectories further back in time simply because they become increasingly uncertain. Given the aerosol could have arrived in the Arctic many weeks or months before we sampled it, back trajectories are of limited use beyond establishing the likelihood of more local sources.

We have a paper on the characterisation of the SEM technique (Sanchez-Marroquin et al., 2020), so we think we do not need to take a lot of space in this paper proving this again.

We think our chemical evaluation of the particles is enough to rule them as "mineral dust" rather than some unspecified industrial aerosol since the industry in the area is mostly oilfields and these would produce other types of aerosol particles.

Lines 247-248: This cannot be confirmed without INP treatments. The authors should reword this to demonstrate this is a speculation, albeit a legitimate one.

We use the word 'suggest' which is already rather soft. We have added 'perhaps' as well: 'Hence, we suggest that the enhanced ice-nucleation ability of our samples is perhaps due to the presence of biological material'

Lines 254-257: Can the authors elaborate on this? What specifically would the biological material be from during this time of year? Should reference papers like Creamean et al. (2022) and Santl-Temkiv et al. (2019) here that do evaluate INPs/biological particles in the spring. Porter was in the late summer, so it is not exactly relevant for the Mar measurements here. Late summer sources can vary quite a bit from spring, given the contrasting transport conditions, surface open water, and marine versus sea ice biological productivity. Need to compare with previous ISDAC results (also a spring flight campaign).

It is very hard to say what it might be. Given our statement that there is some biological ice-nucleating material present is tentative, then we feel a lengthy discussion is not warranted. We have added the additional citations suggested by the referee.

We have added the limited amount of published ISDAC data to figure 3.

Fig 1: The flight IDs are difficult to discern without the table being in the main text.

The table has been moved to the main text.

Fig 2: Starts at -14C...is this a sample limit or instrumental bias? If the former, is this because of the blanks for sample volume?

The background has a tail to these temperatures. The data from this campaign unfortunately suffered from a higher background than other campaigns we have done. However, this is taken into consideration in our background subtraction and error analysis.

Fig 3: "All Arctic" is a misnomer. These are from Porter et al. from one study in Aug/Sep. Why are the authors showing data from other studies in the same timeframe, but from the late summer for Porter? The Porter data are not actually relevant for this figure. If the authors want to show the specific studies indicated in comparison to, truly all Arctic data, then other studies should be included from other times of the year as well.

We have replaced the All Artic range by a new graph (Fig. 3b) which compares our measurements with all the existing literature of INP measurements in the Arctic.

5. Figure 4 and Table 1: I understand that Figure 4 left is the number size distribution. I assume Figure 4 right is the mass fraction of different components. My question is how to calculate the surface area of sea salt and dust in Table 1.

The left figures correspond to number size distribution while the right side fractions correspond to the number fraction of particles in each bin. This has been clarified. An explanation has also been added about the surface area: "The number size distribution is multiplied by the fraction of particles in each category and bin to calculate the number size distribution of each category. Then these number size distributions are turned into surface area size distributions and integrated to obtain the surface area of each category, as shown in Table 2."

6. Line 365 Appendix B: Did you use the mean background value of all handling blank samples and subtract this value? The frozen fraction of handling blank samples shows a larger variation (Figure 2a). How to explain the larger range of backgrounds?

In our experience handling blanks for INP analysis on some campaigns can be variable. It is not completely clear why, but very important to characterise with sufficient numbers of handling blanks from throughout the campaign.

7. Figure B3: Please use the correct legends (solid or hollow) for each sample. For example, legend markers of C091_4 should be hollow.

We could not do that as some samples have both upper limits (hollow markers) and full measurements (filled markers) at the same time. We have clarified it in the caption: "Note that full markers corresponds to measurements above the limit of detection, while hollow markers correspond to upper limits. This has not been specified in the legend as some samples have both upper limits and measurements at the same time"

---

## Author Response (AR2)

We would like to thank the editor and the reviewers for all the work they have done in order to improve the current manuscript. We have addressed all the comments raised during the last round of reviews.